# LiveFigure: Generating Editable Scientific Illustration with VLM Agents

**Chenyang Shao** [1 2] **Jiahe Liu** [1] **Fengli Xu*** [1 2] **Yong Li** [1 2]

## Abstract

Scientific illustrations are essential for depicting conceptual designs, methodologies, and experimental workflows in research, which play a pivotal role in communicating complex academic insights. However, creating high-quality scientific illustrations remains a labor-intensive task for human scientists. While recent generative image models have advanced prompt-based editing, the synthesis of fully **editable** figures remains a fundamental challenge. Valid editability involves structured transformations of graphical elements, scales, attributes, and text, rather than simple pixel-level changes. Existing models generate raster outputs that do not support manual correction or layout adjustment, limiting their utility in scientific publishing, where editable vector figures are typically required for submission. To address this challenge, we introduce **LiveFigure**, an agentic framework driven by VLM agents that imitates the multi-step drawing workflow of human researchers. It first plans figure blueprints by drawing inspiration from high-quality references in previous works, then generates executable scripts that produce figures via the PowerPoint interface based on skills and experience, and finally refines the outputs with targeted visual diagnostics, producing fully vectorized, editable figures that meet publication standards. Extensive experiments demonstrate that LiveFigure generates inherently editable figures, achieving 80% publication-readiness in only 17 manual edits, far surpassing the 24% rate of the strongest baseline, NanoBanana. Human preference studies further validate this advantage, with LiveFigure securing a 60% win rate against NanoBanana. Our code is available at https://github.com/tsinghua-fib-lab/LiveFigure.git.

[1]Department of Electronic Engineering, BNRist, Tsinghua University, Beijing, China [2]Zhongguancun Academy. Correspondence to: Fengli Xu <fenglixu@tsinghua.edu.cn>.

*Proceedings of the 43rd International Conference on Machine Learning*, Seoul, South Korea. PMLR 306, 2026. Copyright 2026 by the author(s).

## 1. Introduction

Scientific illustration figures serve as a crucial medium for illustrating methods, workflows, and conceptual frameworks, enabling readers to quickly grasp the core logic and key ideas underlying a study (Larkin & Simon, 1987; Tufte, 2001). Despite their central role in research communication, producing high-quality figures remains a labor-intensive process (Rougier et al., 2014): it demands both deep domain knowledge and proficiency with professional design tools, often requiring manual construction of each element using software such as Adobe Illustrator (Rolandi et al., 2011). This manual workflow consumes significant time and effort, limiting researchers' productivity and diverting resources away from the generation of novel scientific insights.

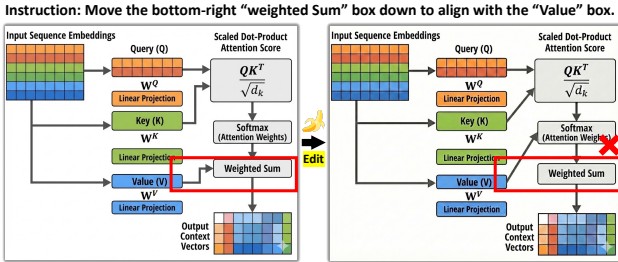

*Figure 1.* Natural language instructions struggle to guide generative models to make precise modifications to scientific figures.

In an effort to reduce the efficiency bottleneck described above, recent advances in generative image models may appear to offer a promising alternative. However, these models fundamentally generate *static, pixel-based raster images*, which are inherently misaligned with the requirements of scientific figures (Li et al., 2025a). On the one hand, this generation paradigm enforces a single-pass, monolithic rendering process, in which figures are treated as indivisible visual outputs rather than compositions of editable semantic elements. Under this paradigm, even minor structural inaccuracies or spelling errors in technical terminology cannot be corrected through localized edits to individual components, connectors, or text elements. Instead, users must rely on iterative natural language instructions that trigger full regeneration of the figure. Even with repeated prompts, precise and fine-grained modifications to individual elements remain largely unattainable, as illustrated in Figure 1. This limitation fails to support the highly iterative, fine-grained, and verifiable editing process required in scientific writing.

On the other hand, top-tier academic venues such as *Nature* and *Science* mandate figures to be submitted in vector formats. Vector graphics are natively editable and support resolution-independent scaling without quality degradation, both of which are essential for accurate scientific communication, high-fidelity printing, and rigorous publication standards. However, existing generative image models are fundamentally incapable of producing vector graphics, making them incompatible with these publication requirements.

In response, the research community has explored various solutions, yet none have fully bridged the gap between raster images and editable scientific figures. AutoFigure (Anonymous, 2025) makes text editable by recognizing and reinserting labels via OCR, but the remaining graphical elements are still pixel-based and cannot be directly modified. Qwen-Image-Layered (Yin et al., 2025) decomposes a single RGB image into a stack of semantically disentangled RGBA layers, allowing users to independently adjust each layer without affecting others. However, each layer still fundamentally relies on pixel-based regeneration, and the method performs poorly when decomposing scientific schematics with complex layouts and numerous elements. Code-based generation methods (Rodriguez et al., 2025; Yang et al., 2024) produce vectorized outputs, but these figures are essentially non-editable and visually simplistic, lacking semantic richness and aesthetic quality.

We argue that fundamentally achieving native editability in scientific figures requires abandoning traditional pixel-based end-to-end image generation methods in favor of a new paradigm, which we term *Cognition-Inspired Procedural Construction*. Observing how human experts create figures reveals that this process is not a matter of pixel manipulation, but a staged cognitive workflow: experts first retrieve relevant reference figures for inspiration, then conceptualize a high-level visual blueprint, next search for or design semantic assets, subsequently assemble these assets using appropriate tools, and finally iteratively refine the figure based on visual feedback. By explicitly modeling and simulating this process, and by driving generation through code rather than pixels, it becomes possible to achieve inherent editability and logical precision.

Building on this paradigm, we propose **LiveFigure**, a multi-agent framework that emulates the cognitive workflow of expert human designers to automatically synthesize natively editable PowerPoint source files, which can be exported as vectorized scientific figures. LiveFigure adopts an agentic pipeline that decomposes the generation into three stages. First, *visual planning via prior induction* distills high-quality schematic design priors from expert-authored figures in top-tier venues, enabling the system to infer reliable layout structures and semantic organization from methodological text, thereby establishing a visually grounded blueprint to guide downstream figure generation. Second, *procedural figure generation via standardized skills and experience enhancement* translates the visual blueprint into executable figure-generation programs. By equipping agents with a library of pre-validated high-level drawing skills and injecting accumulated debugging experience as constraints, LiveFigure enables robust code synthesis that produces fully editable PowerPoint figures. Finally, *targeted refinement via visual diagnostics* closes the loop between code execution and visual perception. Through iterative observation-and-refinement cycles, the system identifies subtle visual defects, such as element occlusion or misalignment, and makes targeted code-level corrections to achieve publication-ready figures.

Extensive experiments demonstrate that LiveFigure produces **fully editable** figures with high visual clarity and aesthetic quality. In human evaluations, 80% of figures generated by LiveFigure were deemed suitable for publication after at most 17 minor edits, substantially outperforming the strongest baseline, NanoBanana, which achieves only 24%. Additionally, human pairwise preference studies show that LiveFigure achieves a 60% win rate against NanoBanana, confirming its superior usability and quality. Our contributions can be summarized as follows:

1. We propose a novel agentic paradigm, *Cognition-Inspired Procedural Construction*, for scientific figure generation, which imitates the staged workflow of human experts and integrates visual planning, skill-augmented procedural generation, and iterative visual refinement.

2. We develop **LiveFigure**, a practical framework that produces fully editable, visually clear, and semantically rich scientific figures, overcoming the limitations of raster-based generative models and simplistic code-based approaches.

3. We validate the practical usability of LiveFigure through extensive experiments and human studies, demonstrating that only minimal manual adjustments are needed for immediate adoption.

**Conflict of Interest Disclosure.** The authors declare that they have no financial conflicts of interest related to this paper.

## 2. Related Works

### 2.1. General Image Generation

Driven by the scaling of Diffusion Transformer architectures and advancements in multimodal logical reasoning, general-purpose image generation models have witnessed exponential progress in recent years. State-of-the-art closed-source models, represented by GPT-Image-1.5 (OpenAI, 2025) and Gemini Nano Banana (DeepMind, 2025), are

now capable of synthesizing photorealistic images and natural scenes with high aesthetic fidelity. Recent pipelines such as AutoFigure (Anonymous, 2025) largely rely on these foundational models to construct scientific imagery. However, within the rigorous context of scientific figure creation, these pixel-based end-to-end models face the fundamental challenge of *uneditability*. Furthermore, issues regarding textual and logical hallucinations persist (Shimoda et al., 2025); distortions in text, topological incoherence, and content fabrication occur frequently (Chen et al., 2024; Li et al., 2025b), failing to meet the stringent standards required for academic publication.

To mitigate these editability constraints, the research community has begun exploring structure-aware generation paradigms. For instance, OmniGen (Xiao et al., 2025) introduces a unified generation framework attempting to address various generation and editing tasks within a single model; yet, its outputs remain in an indecomposable raster format. Similarly, Qwen-Image-Layered (Yin et al., 2025) employs latent space decomposition techniques to generate images as separate RGBA layers, thereby enabling users to move or remove individual objects. While this approach improves object-level controllability to a certain extent, it remains fundamentally a pixel-based operation. The limitations in layer count and separation granularity fall short of supporting the fine-grained text editing and vector-level stylistic adjustments essential for scientific figures.

### 2.2. Autonomous Agents for Scientific Research

Amid the surging wave of "AI for Science", autonomous agents are rapidly evolving into pivotal tools for augmenting human scientific capabilities. By automating laborious components of the research workflow, these agents empower researchers to concentrate on high-level ideation and innovation. In the realm of *information acquisition and synthesis*, systems such as Deep Research (Google, 2025; OpenAI, 2025) have demonstrated exceptional long-context reasoning capabilities, assisting researchers in rapidly extracting and synthesizing critical knowledge from vast repositories of literature. Regarding *domain-specific execution*, tools like ChemCrow (Bran et al., 2023) and Data Interpreter (Hong et al., 2025) have validated the auxiliary value of agents in tasks ranging from chemical synthesis planning to complex code-based data analysis. For *hypothesis verification and optimization*, evolutionary frameworks represented by AlphaEvolve (Novikov et al., 2025) serve as powerful exploration tools, facilitating the automated discovery and optimization of superior model architectures within expansive search spaces. While these systems significantly accelerate knowledge production, a gap remains in the automated dissemination and communication of findings. Most existing assistants focus on *reading*, *calculating*, or *executing* (Shao et al., 2025), leaving the creation of visual schematics to manual labor. LiveFigure addresses this gap by extending the scope of research automation to the visual domain, serving as a specialized agent that transforms abstract research designs into publication-standard, editable figures.

## 3. Methodology

We conceptualize **LiveFigure** as an agentic framework that simulates the cognitive workflow of expert human designers. As illustrated in Figure 2, LiveFigure decomposes the figure generation task into three stages: **Visual Planning via Prior Induction** ($\Psi_{plan}$): aims to uncover high-quality design patterns from top-tier conferences, establishing reliable priors for visual layout in figure-generation tasks. **Procedural Figure Generation via Standardized Skills and Experience Enhancement** ($\Psi_{assemble}$): maps the design blueprint into executable figure-generation scripts, ensuring both code executability and logical consistency. **Targeted Refinement based on Visual Diagnostics** ($\Psi_{refine}$): employs a multi-modal closed-loop feedback mechanism to iteratively optimize generated figures, enabling precise corrections and visual enhancements.

Formally, let $\mathcal{T}_{in}$ denote the input methodological text. Our objective is to synthesize an editable PowerPoint figure $\mathcal{F}_{final}$ that maximizes both semantic alignment and visual clarity. We formulate this generation pipeline as a sequential composition of the three specialized mapping functions defined below:

$$\mathcal{F}_{final} = \Psi_{refine} \circ \Psi_{assemble} \circ \Psi_{plan}(\mathcal{T}_{in}) \quad (1)$$

### 3.1. Visual Planning via Prior Induction (Stage I)

As is often said, the first step is the hardest: generating concrete visual designs solely from input methodological text ($\Psi_{plan} : \mathcal{T}_{in} \to \mathcal{B}$) requires substantial reasoning. In this phase, we introduce a **Visual Prior Induction** mechanism, which distills reusable schematic design priors from high-quality methodological figures in the literature. By leveraging expert-designed exemplars from top-tier conferences, this mechanism establishes reliable visual priors that guide the subsequent blueprint planning process.

To facilitate effective Visual Prior Induction, we construct a **figure-text knowledge base focused on scientific figures**, denoted as $\mathbb{K} = \{(v_i, c_i, d_i)\}_{i=1}^{N}$, where $v_i$ is the figure, $c_i$ is the caption, and $d_i$ represents the dense technical description. The construction of $\mathbb{K}$ involves rigorous structure-aware filtering and context-aware extraction to ensure it contains only high-quality methodological flowcharts and framework schematics (see Appendix A.4 for detailed construction protocols).

Based on this knowledge base, the retrieve agent first fetches the top-$k$ most semantically relevant reference

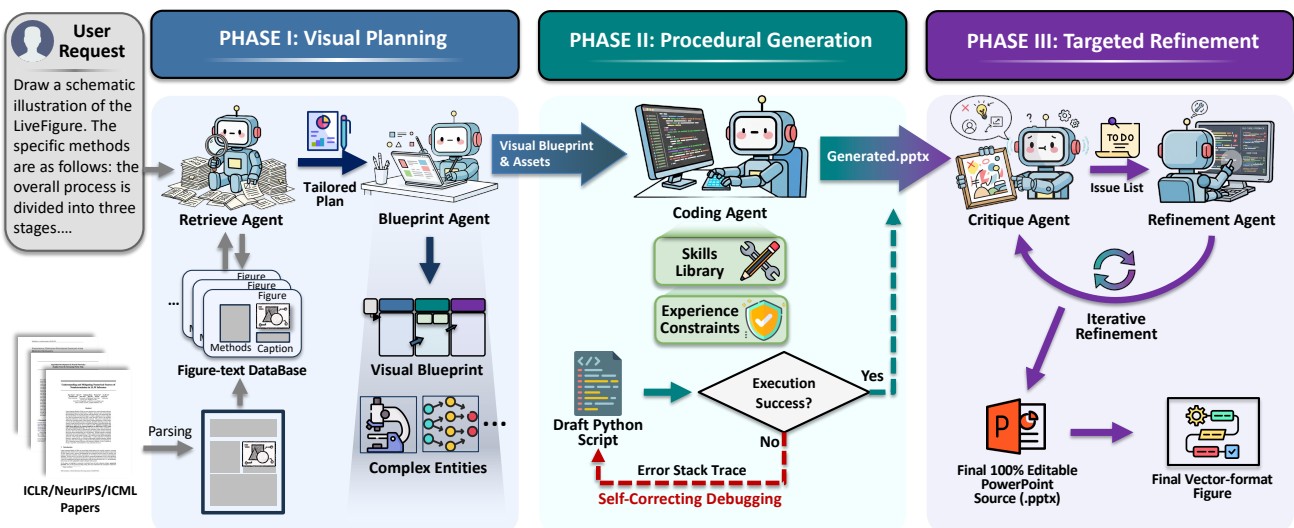

*Figure 2.* Overview of the proposed LiveFigure. The framework simulates human figure design via three stages: (I) Visual Planning via Prior Induction, (II) Procedural Figure Generation via Skills and Experience, and (III) Targeted Refinement via Visual Diagnostics. **This figure itself was also generated by LiveFigure** and further refined through 14 steps of manual human editing.

figures from $\mathbb{K}$, which can be formulated as: $\mathcal{R} = \text{TopK}_{k \in \mathbb{K}} \text{sim}(E(\mathcal{T}_{in}), E(k))$, where $E(\cdot)$ denotes the embedding function (e.g., Qwen3-Embedding). Next, a VLM jointly analyzes the retrieved references $\mathcal{R}$ and the user input $\mathcal{T}_{in}$, examining the layout organization and visual composition embodied in the reference figures. By abstracting and distilling their underlying design principles, the model adapts these visual priors to the target methodology, producing a tailored structural plan $\mathcal{S}_{plan}$ aligned with the input specification: $\mathcal{S}_{plan} = \text{VLM}_{reason}(\mathcal{T}_{in}, \mathcal{R})$. Finally, the blueprint agent takes both the original context and the structural plan as prompts to generate the visual blueprint $\mathcal{B}$, which provides the spatial guidance for the downstream procedural generation:

$$\mathcal{B} = \text{Gen}_{img}(\mathcal{T}_{in}, \mathcal{S}_{plan}). \tag{2}$$

For domain-specific entities $\mathcal{E}_{entity}$ (e.g., "microscope," "robotic arm") mentioned in $\mathcal{S}_{plan}$, whose visual complexity exceeds what can be expressed by basic geometric primitives, an asset generation module further synthesizes style-consistent assets. To achieve high-fidelity generation with minimal inference cost, we employ a grid-based batch generation strategy. Instead of synthesizing assets individually, we prompt the model to generate a composite image containing an $M \times N$ grid of icons in a single pass, strictly conditioned on the visual style of $\mathcal{S}_{plan}$. Formally, we define a grid generation function $\text{Gen}_{grid}$ followed by an automated post-processing operator $\Phi_{post}$ (which performs grid cropping and background removal). The final asset library $\mathbb{A}$ is obtained as:

$$\mathbb{A} = \Phi_{post}\left(\text{Gen}_{grid}(\mathcal{E}_{entity} \mid \mathcal{S}_{plan})\right). \tag{3}$$

This mechanism ensures that complex scientific entities are visually consistent and ready for seamless integration.

### 3.2. Procedural Figure Generation via Skills and Experience (Stage II)

After establishing the visual blueprint $\mathcal{B}$ and the asset library $\mathbb{A}$, the objective of Phase II is to procedurally generate an editable figure. We formalize this process as a mapping $\Psi_{gen} : (\mathcal{B}, \mathbb{A}) \to \mathcal{F}_{init}$, where $\mathcal{F}_{init}$ represents the initial editable PowerPoint figure.

In this work, we choose **Microsoft PowerPoint** as the carrier for editable figures. *Why PowerPoint?* This decision is grounded in a careful consideration of user accessibility and automation feasibility. While professional design tools (e.g., Adobe Illustrator) are the industrial standard for vector graphics, their closed ecosystems and steep learning curves make them poorly suited for AI-driven automation. In contrast, PowerPoint offers a rare combination of broad user accessibility and developer-level openness. On the user side, it serves as a de facto universal language in the research community, ensuring ubiquitous availability. On the developer side, its rich programmatic interfaces (via OpenXML) enable fine-grained manipulation of graphical primitives, while also supporting seamless export to standard vector formats. Thus, PowerPoint uniquely satisfies the dual requirements of low-friction user editing and high-precision model control. A comprehensive rationale for selecting PowerPoint over traditional vector formats like PDF or SVG is detailed in Appendix A.7.

Nevertheless, directly prompting VLMs to generate raw Python code based on `python-pptx` for procedural figure generation remains non-trivial. In practice, this approach frequently suffers from excessively verbose code, API hallucinations, abundant implementation bugs, and suboptimal spatial reasoning. To address these challenges, we propose

a **Standardized Skill-based Generation** paradigm, augmented with **Experience Enhancement**.

**Standardized Skills as Atomic Primitives.** We construct a Python library of *Standardized Skills*, denoted as $\mathbb{S} = \{s_1, s_2, \ldots, s_M\}$. Each skill $s_j$ is pre-debugged and rigorously validated, and encapsulates complex rendering logic, such as connector routing and nested text-shape composition, into high-level semantic interfaces. The procedural generation process $\mathcal{M}_{gen}$ is modeled as synthesizing an executable Python script $\mathcal{C}_{init}$ conditioned on the visual blueprint $\mathcal{B}$ and the skill library $\mathbb{S}$. The initial editable figure $\mathcal{F}_{init}$ is then obtained by executing this script in the Python environment:

$$\mathcal{F}_{init} = \text{Exec}(\mathcal{C}_{init}), \quad \mathcal{C}_{init} \sim \mathcal{M}_{gen}(\cdot \mid \mathcal{B}, \mathbb{S}). \quad (4)$$

This abstraction of skills serves two critical purposes:

1. First, it **guarantees executability**. By constraining the model's action space to a curated set of verified atomic functions, we substantially reduce syntax errors and API hallucinations, thereby improving the executability of generated code and the overall system efficiency.
2. Second, it enables **cognitive offloading**. The abstraction decouples high-level layout reasoning from low-level implementation details. The model can focus on semantic layout decisions, such as "component A should be connected to component B with a polyline arrow", without explicitly computing anchor points or intermediate pixel-level routing coordinates. Moreover, the encapsulation provided by standardized skills significantly reduces code length and complexity, where a single skill invocation can replace dozens of lines of intricate layout logic.

**Experience-Driven Constraint Injection.** When interacting with visualization and rendering libraries, LMs often produce API hallucinations or invalid parameter combinations due to biases in their training data. To systematically mitigate such errors, we introduce an *Evolving Experience Injection* mechanism, which distills debugging experiences accumulated during development and large-scale generation into formalized negative constraints $\mathbb{E}_{neg}$.

As the system processes an increasing number of generation cases, the experience repository is continuously updated by capturing newly observed runtime errors. For instance, to prevent a known API misuse, prohibitive rules are automatically incorporated into the prompt, such as: *"NEVER use* `slide.shapes.add_shape(MSO_SHAPE.LINE, ...)` *directly; use* `add_connector` *instead."* By progressively integrating domain-specific failure knowledge in this automated manner, the mechanism performs pre-pruning over the generation search space, ensuring that the model learns from past mistakes and blocks known erroneous pathways at the level of generation logic itself.

**Self-Correcting Execution Loop.** While standardized skills and experience constraints significantly reduce error rates, to handle sporadic complex logic conflicts, we incorporate a runtime feedback-based iterative debugging mechanism. Upon execution failure, the system captures the error stack trace $\epsilon$ and feeds it back to the model. Let $\mathcal{C}_{draft}$ denote the initial code generated. We define the debugging iteration sequence $\{\mathcal{C}_{debug}^{(t)}\}$ initialized with $\mathcal{C}_{debug}^{(0)} = \mathcal{C}_{draft}$:

$$\mathcal{C}_{debug}^{(t+1)} = \text{Refine}(\mathcal{C}_{debug}^{(t)}, \epsilon^{(t)}), \quad \text{s.t. } t < T_{max}. \quad (5)$$

This loop terminates upon successful execution, yielding the validated executable script $\mathcal{C}_{exec}$. This mechanism serves as a final safety net, ensuring the system can autonomously recover from unforeseen runtime errors.

### 3.3. Targeted Refinement via Visual Diagnostics (Stage III)

Although the script $\mathcal{C}_{exec}$ obtained from Phase II is valid and executable, the resulting figure $\mathcal{F}_{init}$ often exhibits subtle visual defects invisible to code-level logic, such as element occlusion or inconsistent styling. To bridge the gap between code logic and visual perception, we design a **Visual Diagnosis-Driven Refinement** closed-loop, formally modeled as an optimization mapping $\Psi_{refine} : \mathcal{F}_{init} \to \mathcal{F}_{final}$.

We formulate this phase as an iterative "observe-diagnose-refine" process. Let $\mathcal{C}^{(0)} = \mathcal{C}_{exec}$ denote the starting script inherited from Phase II. At each iteration $k$, the system renders the current script into a visual snapshot $I^{(k)}$. A VLM acts as a "visual critic" to perform diagnosis, outputting a structured **Actionable Issue List** $\mathcal{L}_{issue}$ that precisely localizes specific flaws (e.g., "Text box A overlaps with Arrow B"): $\mathcal{L}_{issue}^{(k)} = \text{VLM}_{critic}(I^{(k)})$. Subsequently, the agent executes targeted refinement. Instead of regenerating the entire script, it applies incremental updates to the code conditioned on the feedback list: $\mathcal{C}^{(k+1)} = \text{Refine}(\mathcal{C}^{(k)}, \mathcal{L}_{issue}^{(k)})$. This loop continues until the issue list is empty or convergence is reached. The final publication-ready figure is obtained as $\mathcal{F}_{final} = \text{Exec}(\mathcal{C}_{final})$. This mechanism ensures that the output meets stringent aesthetic standards through human-mimetic visual feedback. The prompt templates used for procedural generation and visual critique are provided in Appendix C.3.

## 4. Experiments

### 4.1. Experimental Setup

**Baselines:** We compare our method against several representative image generation methods. Specifically, we include **Gemini-3-Pro-Image (Nano Banana)**, a commercial image generation model released by Google in November 2025; **GPT-Image-1.5**, OpenAI's latest general-purpose

image generation model introduced in December 2025; **Imagen-4.0-Ultra**, a high-fidelity text-to-image model from Google's Imagen 4 family made generally available in 2025; **Qwen-Image**, an open-source image generation model released by Alibaba in August 2025; **Grok-2-Image**, an industry image generation system developed by xAI and released in August 2024; and several widely-used programmatic figure generation libraries, including **Mermaid**, **Graphviz**, **Matplotlib**, **TikZ-V1**, and **HTML/CSS**. Because AutoFigure has not been open-sourced, it is not included in our baseline comparisons. For more detailed information on these baselines, please refer to Appendix C.1. Our framework uses Gemini-3-Pro and Gemini-3-Image-Pro as the backbone models for the agents.

**Edit Distance Metric:** We quantify practical editability using **Edit Distance**, defined as the number of atomic modification steps a human user performs to finalize a generated figure. Since PPTX files are internally represented as structured XML trees, we compute this metric by directly diffing the underlying XML representations of the figure before and after user editing. Specifically, we count three types of operations: *Modify*, where an element with the same ID exists in both versions but differs in attributes such as position, color, or text; *Add*, where an element appears only in the edited version; and *Delete*, where an element is removed. This design ensures that repeated adjustments to the same entity are aggregated into a single operation, yielding an objective and robust measure of human editing effort.

**Visual Evaluation Metrics:** We evaluate each figure along three visual dimensions, comprising a total of nine metrics. All metrics are higher-is-better ($\uparrow$).

- **Visual Design**: This dimension evaluates the visual quality of the figure. It includes **Aesthetic Quality**, which measures whether the figure exhibits modern scientific illustration aesthetics in color, layout, and whitespace while maintaining a professional appearance; **Visual Expressiveness**, which measures whether abstract concepts are effectively transformed into intuitive visual symbols through icons and visual metaphors; and **Professional Polish**, which measures whether alignment, spacing, connectors, and fonts reach publication-quality standards.
- **Information Clarity**: This dimension evaluates the effectiveness of information communication. It includes **Clarity**, which measures whether readers can quickly identify the core information and structural focus of the figure; **Logical Flow**, which measures whether layout and arrows guide the reader along a causally consistent sequence; and **Text Legibility**, which measures whether text is clear, readable, correctly spelled, and free from generative errors.
- **Content Fidelity**: This dimension evaluates how faithfully the figure reflects the underlying content. It includes **Accuracy**, which measures whether nodes, connections, and information flow accurately reflect the original scientific description; **Completeness**, which measures whether all key modules, steps, or essential information mentioned in the text are represented in the figure; and **Appropriateness**, which measures whether the abstraction level and visual style of the figure are suitable for the target audience and academic context.

**Dataset:** Our evaluation is conducted on a curated dataset of scientific schematic figures derived from accepted papers at ICLR 2024, NeurIPS 2024, and ICML 2024. For each conference, we automatically extract 100 figure–text pairs, resulting in a total of 300 test instances. Each pair consists of a scientific schematic and its associated textual description, including the figure caption and relevant method descriptions. Appendix A.3 details the domain distribution of these 300 papers. Further details of the dataset construction are provided in the Appendix A.6.

### 4.2. Main Results

To quantify practical editability and utility, we measure Edit Distance—the number of atomic modification steps required to finalize a figure. As illustrated in Figure 3, our full model (red) demonstrates superior efficiency, achieving a median (50%) adoption rate at just 8 steps and reaching the 80% threshold at 17 steps. The dense concentration of data points in the low-edit-distance region ($x < 10$) confirms that the majority of our generated figures require only minimal refinement to meet user standards.

The necessity of our hierarchical design is further evidenced by the ablation studies. Removing the Skill Library (w/o Skills) causes the most severe performance drop, increasing the median edit distance to 31 steps (nearly $4\times$ the effort), which indicates a struggle with syntactic precision without executable tools. Similarly, removing the Experience Module (w/o Experience) delays median adoption to 20 steps, demonstrating the value of injecting accumulated debugging experience and failure-derived constraints into the generation process. In contrast, general-purpose VLMs struggle significantly with zero-shot generation. Nano Banana and gpt-image-1.5 achieve adoption rates of only 24% and 15%, respectively. This sharp contrast underscores the limitations of generative image models in specialized scientific plotting and the necessity of our framework.

### 4.3. Visual Quality and Content Fidelity Evaluation

Table 1 compares LiveFigure with raster generative models and code-based tools across three evaluation dimensions, under two input settings: **V1**, which uses caption-only input, and **V2**, which provides both the figure caption and the corresponding method description. LiveFigure achieves a clear advantage over traditional programmable tools such

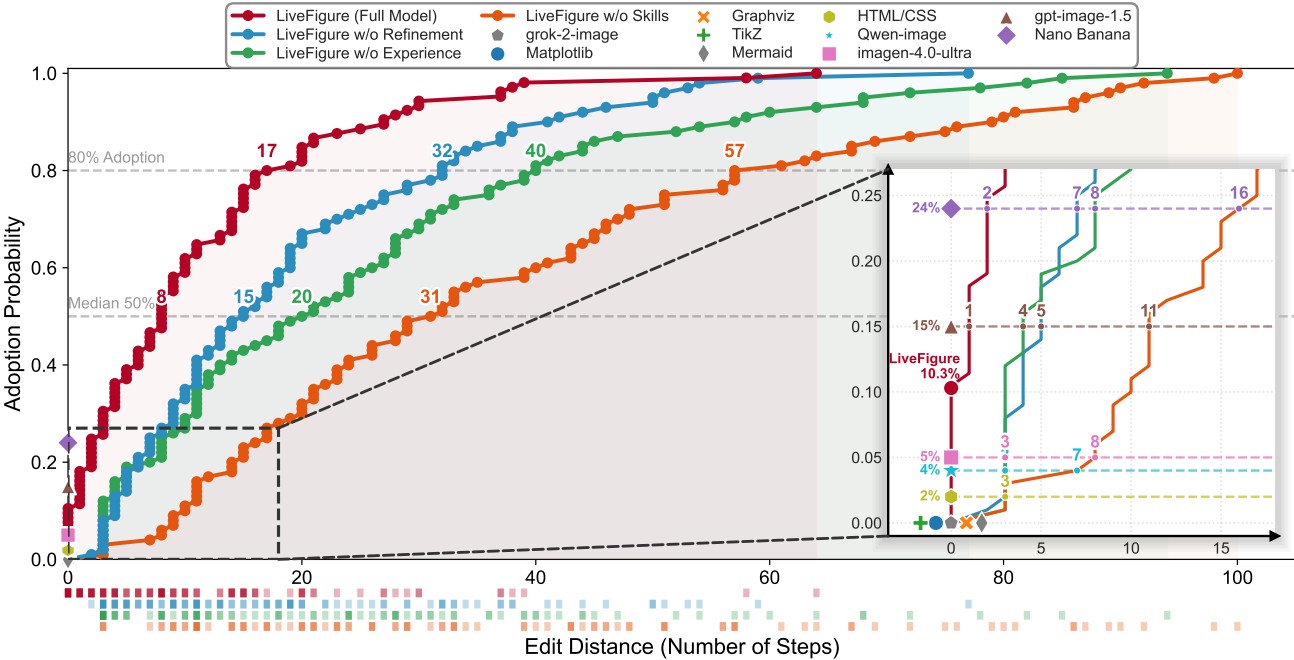

*Figure 3.* Cumulative adoption probability curves with respect to edit effort. The x-axis represents the Edit Distance required for user adoption. The y-axis shows the cumulative probability of a generated figure being adopted. The main plot compares our full method (red) against three ablation variants. The right zoom panel details the modification-free adoption rates ($x = 0$) for all methods, with horizontal dashed lines projecting non-zero baselines to their equivalent step count on our method's curve. Horizontal dashed lines in the main plot mark median (50%) and 80% adoption thresholds. Rug plots at the bottom visualize the density distribution of edit distances.

| Models | Visual Design↑ | | | Information Clarity↑ | | | Content Fidelity↑ | | | Ave.↑ |
|---|---|---|---|---|---|---|---|---|---|---|
| | Aesthetic | Express. | Polish | Clarity | Flow | Legibility | Accuracy | Complete. | Appropriate. | |
| **V1: Caption-only Input** | | | | | | | | | | |
| Nano Banana-V1 | 7.68 | 6.73 | 7.81 | 7.14 | 7.49 | **8.54** | 5.92 | 6.20 | 7.12 | 7.1811 |
| gpt-image-1.5-V1 | **8.08** | **7.05** | **8.09** | **7.44** | **7.63** | 8.21 | 5.75 | 6.02 | **7.31** | 7.2867 |
| Qwen-image-V1 | 8.00 | 6.83 | 7.14 | 5.49 | 5.44 | 5.88 | 3.90 | 4.28 | 5.41 | 5.8189 |
| grok-2-image-V1 | 5.90 | 4.54 | 5.30 | 3.66 | 3.62 | 2.91 | 3.18 | 3.26 | 3.68 | 4.0056 |
| imagen-4.0-ultra-V1 | 7.63 | 6.31 | 7.33 | 5.58 | 5.80 | 6.23 | 4.65 | 4.61 | 5.70 | 5.9822 |
| Mermaid-V1 | 2.49 | 1.70 | 1.78 | 3.01 | 3.18 | 3.41 | 4.35 | 2.12 | 2.12 | 2.6844 |
| Graphviz-V1 | 3.53 | 3.52 | 4.21 | 5.21 | 6.24 | 7.34 | 5.83 | 6.03 | 6.59 | 5.3889 |
| Matplotlib-V1 | 5.99 | 5.22 | 5.82 | 5.66 | 5.88 | 6.70 | 4.80 | 4.92 | 5.53 | 5.6133 |
| TikZ-V1 | 3.59 | 2.95 | 3.51 | 3.12 | 3.28 | 3.87 | 2.62 | 2.74 | 3.17 | 3.2056 |
| HTML/CSS-V1 | 6.76 | 5.87 | 7.09 | 6.64 | 6.80 | 7.34 | 5.61 | 5.86 | 6.52 | 6.4989 |
| LiveFigure-V1 | 7.69 | 6.86 | 7.76 | 7.07 | 7.29 | 8.29 | **6.80** | **6.69** | 7.26 | **7.3011** |
| **V2: Caption + Method Description Input** | | | | | | | | | | |
| Nano Banana-V2 | **7.94** | **7.16** | 7.99 | 7.57 | **8.17** | **8.65** | 7.61 | 7.71 | **8.07** | 7.8744 |
| gpt-image-1.5-V2 | 7.91 | 7.05 | **8.07** | **7.65** | 7.98 | 8.60 | 7.06 | 7.12 | 7.34 | 7.6422 |
| Qwen-image-V2 | 7.43 | 6.42 | 6.25 | 5.43 | 5.84 | 5.43 | 4.75 | 5.22 | 5.85 | 5.8467 |
| grok-2-image-V2 | 5.53 | 4.44 | 4.42 | 3.39 | 3.78 | 2.97 | 3.15 | 3.45 | 3.61 | 3.8600 |
| imagen-4.0-ultra-V2 | 7.89 | 6.41 | 7.29 | 5.45 | 5.59 | 6.56 | 4.91 | 5.28 | 6.02 | 6.1556 |
| Mermaid-V2 | 2.98 | 1.97 | 1.70 | 3.24 | 3.02 | 3.58 | 4.40 | 2.77 | 2.99 | 2.9611 |
| Graphviz-V2 | 4.28 | 3.87 | 4.39 | 5.65 | 6.43 | 7.60 | 6.34 | 6.81 | 6.61 | 5.7756 |
| Matplotlib-V2 | 5.75 | 5.07 | 5.42 | 5.07 | 5.72 | 5.70 | 5.91 | 6.09 | 5.76 | 5.6100 |
| TikZ-V2 | 3.53 | 2.93 | 3.53 | 2.93 | 3.35 | 3.87 | 3.05 | 3.15 | 3.33 | 3.2967 |
| HTML/CSS-V2 | 6.60 | 5.76 | 7.01 | 6.38 | 6.76 | 7.29 | 6.39 | 6.68 | 6.72 | 6.6211 |
| LiveFigure-V2 | 7.87 | 7.08 | 7.87 | 7.59 | 7.75 | 8.62 | **8.29** | **8.16** | 7.79 | **7.8911** |

*Table 1.* VLM-based evaluation results across visual design, information clarity, and content fidelity dimensions. The **best** and second best values in each column (within V1 and V2 groups) are highlighted. Ave. denotes the average over nine metrics across three visual dimensions. Higher values indicate better performance for all metrics↑.

| Model Variant | Executability (↑) | Debug Turns (↓) | VLM Score (↑) |
|---|---|---|---|
| w/o Experience | 40.0% | 3.3 | 6.52 |
| w/o Skills | 77.3% | 1.6 | 5.47 |
| w/o Refinement | 83.5% | 0.46 | 6.78 |
| w/o Visual Prior | 82.3% | 0.55 | 6.81 |
| Full LiveFigure | 83.5% | 0.53 | 7.28 |

*Table 2.* Ablation study on the Experience, Skills, Refinement and Visual Prior modules.

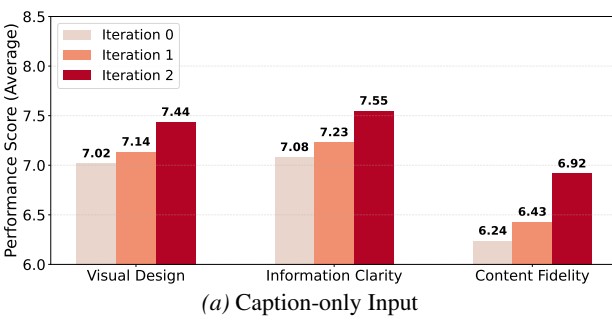

*(a)* Caption-only Input

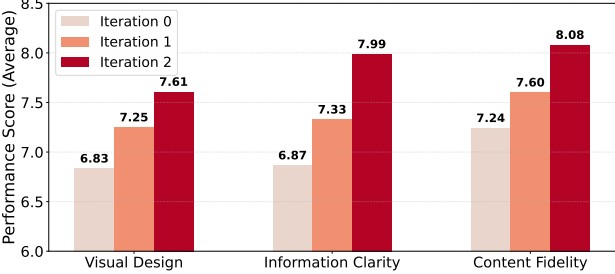

*(b)* Caption + Method Description Input

*Figure 4.* Performance comparison across iterations. Metrics are aggregated based on 3 dimensions of visual evaluation.

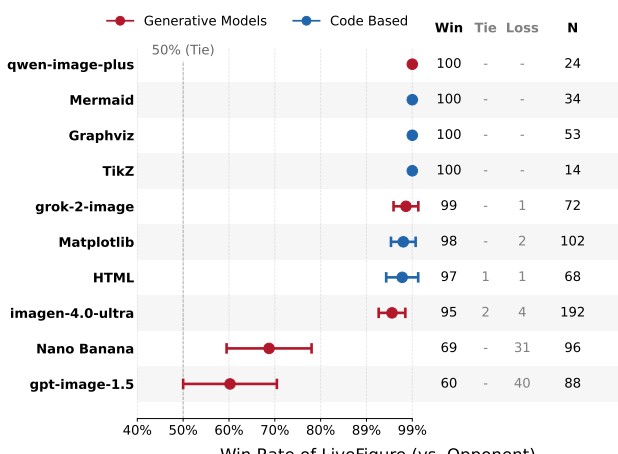

*Figure 5.* Head-to-head human preference evaluation. We compare LiveFigure against baselines under a double-blind, pairwise comparison. The forest plot displays the adjusted win rate of our model. Error bars indicate 95% confidence intervals. The table on the right details the specific breakdown of Wins, Ties, and Losses for each comparison.

### 4.4. Ablation Study

To validate the contribution of each architectural component, we conducted an ablation study (Table 2) and analyzed the performance dynamics across iterative refinement stages (Figure 4). The results confirm the distinct roles of our proposed modules: the Experience module is foundational for syntactic correctness, as its removal precipitates a catastrophic drop in Executability (40.0%); the Standardized Skills are critical for aesthetic quality, evidenced by the lowest VLM score (5.47) when absent; and the Feedback mechanism serves as a necessary corrective layer. Complementing this, the performance trajectory in Figure 4 demonstrates a consistent monotonic improvement across all dimensions, particularly in Information Clarity (rising from 6.87 to 7.99 in V2), confirming that our targeted refinement based on visual diagnostics can effectively improve visual organization and structural fidelity, leading to high-quality scientific visualizations.

Additionally, a progressive, stage-wise evaluation elucidating the incremental performance gains as data flows through the pipeline is available in Appendix B.2.

### 4.5. Human Evaluation

To assess visual quality and layout correctness, we conducted a double-blind, pairwise human preference evaluation (Figure 5). Evaluators were presented with randomized image pairs and instructed to select the one offering better information accuracy and visual clarity. As shown in Figure 5, our model demonstrates statistically significant superiority over all baselines. Against the code-based methods (e.g., Matplotlib, TikZ), we achieved near-perfect adjusted win rates (> 97%), verifying our advantage in automating

as Graphviz (5.78) and Matplotlib (5.61), whose rigid layouts lead to consistently low Visual Design scores, while our method effectively balances structured editability with high aesthetic quality, reaching an average score of 7.89 in the V2 setting. When compared with SOTA raster generators (e.g., Nano Banana and gpt-image-1.5), LiveFigure remains highly competitive: although raster models slightly outperform in aesthetic appearance, LiveFigure shows clear strengths in Content Fidelity. In particular, under V2 input, LiveFigure attains higher Accuracy (8.29) and Completeness (8.16) than gpt-image-1.5 (7.06 and 7.12), indicating more faithful adherence to scientific structure and logic than pure pixel-based generation. Moreover, performance consistently improves from V1 (7.30) to V2 (7.89), reflecting that richer textual input better supports the generation of structurally correct and scientifically rigorous schematics.

To further quantify how faithfully the generated figures convey the user's initial input, a dedicated evaluation of Intent Conveyance Scores is presented in Appendix B.1.

complex styling. Furthermore, our method outperforms state-of-the-art generative models, securing win rates of 69% against Nano Banana and 60% against gpt-image-1.5, with all 95% confidence intervals strictly exceeding the 50% threshold. Appendix C.2 details the GUI design used for this experiment.

Comprehensive details regarding the expert panel's background and the statistical reliability of these results are documented in Appendix A.1.

### 4.6. Qualitative Analysis and Case Studies

To further demonstrate LiveFigure's capabilities across diverse scenarios, we provide extensive qualitative case studies in the supplementary material. Specifically, Appendices B.3 and B.4 present cross-model comparisons on complex structural layouts. Appendix B.5 visually verifies the object-level editability of our native outputs, while Appendix B.6 illustrates the framework's cross-disciplinary generalization through a biological workflow schematic. Finally, we analyze typical generation failure modes in Appendix B.7 and showcase the system's precise natural language-based interactive modification capabilities in Appendix B.8.

## 5. Conclusion

In this work, we introduce LiveFigure, a framework that transforms scientific visualization from static pixel generation to editable object orchestration. By choosing Power-Point as the carrier, we effectively democratize high-quality figure creation, lowering the technical barrier and enabling seamless human-AI collaboration for researchers across disciplines. Looking forward, we aim to extend this paradigm to support multi-modal inputs (e.g., hand-drawn sketches) and broader scientific domains, paving the way for a fully autonomous and interactive research assistant.

## Impact Statements

This paper presents work whose goal is to advance the field of machine learning. There are many potential societal consequences of our work, none of which we feel must be specifically highlighted here.

## Acknowledgements

This work was supported in part by the National Natural Science Foundation of China (Grant No. 62472241), in part by the joint project of Infinigence AI and Tsinghua University, and in part by the Zhongguancun Academy (Grant No. C20250401).

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

# A. Supplementary Details and Discussion

## A.1. Details of Human Evaluation and Expert Background

To rigorously assess the practical utility and publication readiness of the generated illustrations, we conducted a double-blind human preference evaluation. The evaluation panel consisted of seven AI researchers with PhD degrees. Each evaluator has a proven track record, having successfully published at least three papers in top-tier AI conferences or journals.

Given that our test set comprises figures extracted from accepted papers at leading AI venues, these evaluators represent not only the target demographic of our system but also frequent creators of such scientific diagrams. This perfect alignment between the evaluators' area of expertise and the test data ensures that their assessments authentically reflect rigorous, professional academic standards. Furthermore, the winning rates observed in the double-blind preference tests strictly passed the 95% confidence interval test (as shown in Figure 5), demonstrating that the human evaluation results are highly significant and statistically reliable.

## A.2. Cost Analysis

We conducted a detailed cost analysis of the LiveFigure framework based on the generation statistics from our test set. On average, the end-to-end generation of a single scientific figure requires approximately 3 minutes of processing time and incurs a total cost of approximately $0.80. This total cost comprises two main components:

- **Image Generation Model:** Averaging 2 API calls per figure. This consumes approximately 2,000 input tokens (at $2.00/1M tokens) and 3,150 output tokens (at $120.00/1M tokens), which amounts to a cost of $0.38.
- **Vision-Language Model (VLM):** Averaging 12 API calls per figure. This consumes approximately 150,000 input tokens (at $2.00/1M tokens) and 10,000 output tokens (at $12.00/1M tokens), which amounts to a cost of $0.42.

To clarify the incremental token burden introduced by the iterative generation process, we further decompose the costs associated with the self-correction and visual feedback loops:

- **Base Cost (Single-pass):** A successful single-pass generation, without any self-correction or visual refinement, requires 2 image generation calls and 6 base VLM calls. The foundational cost for this single-pass execution is $0.59 ($0.38 for image generation and $0.21 for the VLM).
- **Self-Correction Cost:** When code execution fails and triggers the debug loop, the agent repairs the code based on error logs. Each additional round of self-correction requires 1 extra VLM call, adding a marginal cost of $0.035.
- **Refinement Cost:** During the closed-loop visual refinement stage, the agent fine-tunes the layout and elements based on visual feedback. Each additional round of visual refinement requires 2 extra VLM calls, adding a marginal cost of $0.07.

The iterative loops for self-correction and visual refinement yield significant improvements in both code executability and the final visual quality of the illustrations. When compared to the hours of manual drafting typically required by researchers using professional graphic design software, this marginal computational cost is highly affordable and justified.

## A.3. Domain Distribution of the Test Set

To ensure the diversity and generalizability of our evaluation, the constructed test set comprises 300 papers spanning a wide array of contemporary AI research domains. Table 3 provides a detailed quantitative breakdown of this domain distribution.

The statistical analysis demonstrates that our test set provides broad and balanced coverage across the core fields of modern AI. This comprehensive distribution ensures that the evaluated illustrations accurately represent the highly diverse visual requirements and complex topologies encountered in real-world academic publications.

## A.4. Details of Knowledge Base Construction

The quality of the **Methodological Schematic Knowledge Base** ($\mathbb{K}$) is paramount for the performance of the Visual Deep Research module. We constructed $\mathbb{K}$ using a three-stage pipeline focused on filtering, extraction, and indexing.

| AI Domain | # | % | Core Research Directions & Examples |
|---|---|---|---|
| Foundation Theories, Optimization & Graph Learning | 84 | 28.0% | Fairness, Federated Learning, Graph Neural Networks (GNN), Causal Inference, Topological Data Analysis, Optimization. |
| AI for Science | 72 | 24.0% | Protein Structure Prediction, Drug Discovery, Genomics, Climate Modeling, Medical Image Analysis. |
| Natural Language Processing (NLP) | 51 | 17.0% | LLM Inference, Long Chain-of-Thought Reasoning, Instruction Tuning, Watermarking, Long-context Processing, Evaluation. |
| Computer Vision (CV) | 38 | 12.7% | 3D/4D Reconstruction (Gaussian Splatting), Video Generation, Image Inpainting, Autonomous Driving Perception. |
| Reinforcement Learning & Robotics | 27 | 9.0% | Robotic Manipulation, Offline RL, Multi-Agent Collaboration, Motion Planning. |
| Multimodal Learning | 17 | 5.7% | Vision-Language Alignment, Audio-Language Modeling, Hallucination Mitigation. |
| Systems, Efficiency & Hardware Optimization | 11 | 3.7% | LLM Deployment (Heterogeneous GPU), Operator Linearization, Memory Optimization. |

*Table 3.* Domain distribution of the 300 papers in the test set.

1. **Structure-Aware Filtering:** We collected accepted papers from top-tier conferences (ICLR 2025, NeurIPS 2025, and ICML 2025) and applied a structure-aware filtering process to isolate scientific schematics. To distinguish methodological diagrams from data visualizations, we employed GPT-4o as a "visual reviewer." Through **negative-constraint prompting**, the model was instructed to explicitly exclude experimental figures (e.g., bar charts, line plots) and natural images. Only figures verified as methodological schematics or algorithmic flowcharts, those that explicitly represent structural components and data flows, were retained.

2. **Context-Aware Description Extraction:** Conventional figure-text pairs often rely solely on short captions, which are insufficient to capture complex reproducibility logic. To address this, we designed a two-stage extraction pipeline: first, figure labels in the paper text are identified using regular expressions; next, an LLM (GPT-5-mini) extracts detailed technical descriptions from the surrounding textual context. This **context-aware** mechanism successfully recovers module interactions and data-flow details that are often missing from captions alone.

3. **Dual-Strategy Hybrid Indexing:** To balance retrieval breadth and precision, we employ the Qwen3-Embedding-8B model to construct a dual vector index. The *Caption-Index* is built solely on figure captions, suitable for matching explicit keyword queries, while the *Hybrid-Index* is primarily constructed from long-form descriptions, with fallback to captions when extraction fails, to capture deep semantic and structural similarities. During retrieval, the system dynamically selects between the two indices, ensuring accurate recall of high-quality reference exemplars.

The entire collection and construction process of the image-text pair knowledge base is **fully automated**. Custom parsing scripts automatically extract schematic images, captions, and their surrounding contextual text from PDF documents, while VLMs are subsequently utilized for quality and relevance filtering, **requiring no manual annotation or human intervention**. This automated mechanism ensures that LiveFigure can continuously scale its knowledge base with newly published papers at a marginal cost, persistently enhancing the tool's performance over time.

### A.5. Criteria for Determining "Publication Readiness"

In this study, the concept of "publication readiness" is strictly operationalized through two quantifiable dimensions: expert human evaluation and a structured VLM assessment anchored to official academic publishing guidelines.

### 1. Human Evaluation via Edit Distance

The first dimension quantifies the human effort required to elevate a generated draft to a publishable state. In our evaluation protocol, participants were tasked with manually editing the initial PPTX files generated by LiveFigure. The strict stopping condition for this editing process was the expert's subjective confirmation that the figure had reached a quality standard suitable for direct insertion into a top-tier academic paper.

The evaluation panel consisted of seven PhD researchers, each with a proven track record of at least three accepted papers

in top-tier AI conferences or journals. In their daily research routines, these experts serve as both frequent creators of high-level scientific illustrations and rigorous peer reviewers. Consequently, their consensus on whether a figure meets publication standards is highly professional, reliable, and authentically reflects the current quality consensus within the academic community. The calculated "Edit Distance" thus objectively quantifies the exact number of manual intervention steps necessary to transform a model's initial draft into a final "publication-ready" state.

## 2. VLM-as-a-Judge Evaluation

To complement the human evaluation, we developed an automated VLM-as-a-judge protocol. We rigorously mapped the official figure preparation guidelines from top-tier venues (e.g., Nature, IEEE, NeurIPS) into the three core dimensions and nine quantitative metrics evaluated by the VLM. By assigning the model the persona of a "Senior Scientific Reviewer" in the system prompt, we established evaluation criteria firmly rooted in the following real-world publishing standards:

- **Dimension 1: Visual Design Excellence.** Grounded in the official *Nature* formatting guidelines [1, 2], which strictly prohibit "overlapping text" and "superfluous icons and decorative elements," as well as the *IEEE* author guidelines [5] that emphasize the need to "maintain consistent spacing." Accordingly, metrics such as "Professional Polish" are designed to strictly penalize any boundary clipping, element occlusion, or chaotic spatial layouts.
- **Dimension 2: Communication Effectiveness.** Based on the official *NeurIPS* formatting instructions [4], which mandate that "all artwork must be neat, clean, and legible." To reflect this, our metrics for "Text Legibility" and "Logical Flow" rigorously assess the visual clarity of textual information and the unambiguity of routing connections.
- **Dimension 3: Content Fidelity.** This dimension enforces the absolute baseline set by *Springer Nature*'s image integrity and publishing ethics policies [3], which demand the "accurate and without fabrication" presentation of research results. Within the generative AI context, our "Accuracy" metric imposes severe penalties on any "hallucinated relationships," fabricated visual entities, or erroneous topological logic that deviates from the provided textual input.

By translating these real-world academic publishing norms into a structured scoring rubric, we endow the VLM judge with an authentic expert perspective, thereby achieving a highly standardized and objective quantitative assessment of "publication readiness."

## References for Evaluation Guidelines

[1] Nature. *Final submission artwork guidelines*. Available at: https://www.nature.com/nature/for-authors/final-submission

[2] Nature Portfolio. *Nature Research Figure Guide*. Available at: https://research-figure-guide.nature.com/

[3] Nature Portfolio. *Image Integrity and Standards*. Available at: https://www.nature.com/nature-portfolio/editorial-policies/image-integrity

[4] NeurIPS. *Paper formatting guidelines*. Available at: https://media.neurips.cc/Conferences/NeurIPS2023/Styles/neurips_2023.pdf

[5] IEEE Author Center. *Create Graphics for Your Article*. Available at: https://journals.ieeeauthorcenter.ieee.org/create-your-ieee-journal-article/create-graphics-for-your-article/

## A.6. Details of Test Set Construction

The construction of our evaluation dataset closely follows the same pipeline as the Knowledge Base described above. We collect accepted papers from ICLR 2024, NeurIPS 2024, and ICML 2024, and parse all candidate figures and surrounding text directly from the original PDF files. Given that conference papers contain a large proportion of data visualizations that are unsuitable for schematic generation tasks, a dedicated semantic filtering stage is applied to distinguish methodological schematics from plots and charts.

Specifically, we employ GPT-4o as an automated visual reviewer to inspect each extracted figure. Through constraint-driven prompting, the model is instructed to explicitly exclude experimental result figures, including bar charts, line plots, scatter

plots, and other quantitative visualizations, as well as natural images. Only figures that depict methodological structures, algorithmic workflows, or system-level pipelines, where visual elements represent functional modules and their interactions, are retained. This filtering process ensures that the resulting dataset focuses exclusively on scientific schematics that require structured layout, symbolic abstraction, and precise visual reasoning.

For each retained schematic, we construct a corresponding text description by combining the original figure caption with additional method-level context extracted from the paper body. Rather than relying solely on captions, which are often terse and underspecified, we identify figure references within the method section and use GPT-5-mini to extract the surrounding explanatory text. This context-aware extraction recovers critical information such as module roles, data flow directions, and hierarchical relationships that are essential for faithful schematic reconstruction.

Following this pipeline, we randomly sample 100 schematic–text pairs from each conference, yielding a total of 300 test instances. The resulting dataset serves as a challenging and realistic benchmark for evaluating scientific schematic generation, visual fidelity, and practical editability under human-in-the-loop settings.

### A.7. Why PowerPoint? Rationale for Format Selection

We conducted a systematic evaluation of potential visualization mediums for the LiveFigure system. While PDF (via LaTeX/TikZ) and SVG (via Adobe Illustrator) are common vector formats in academia, we selected PowerPoint (`.pptx`) as the core medium. This decision is driven by considerations of user technical background, software ecosystem compatibility, and the scientific workflow. The rationale is fourfold:

1. **Widespread Accessibility and Low Technical Barrier.** Post-generation refinement is a critical step in scientific visualization. **Availability and Cost:** Professional vector tools like Adobe Illustrator impose high licensing costs and steep learning curves. Similarly, while LaTeX (TikZ) offers precision, its non-WYSIWYG nature restricts the ability of researchers without a CS background to perform fine-grained adjustments. **Skill Alignment:** In contrast, PowerPoint is ubiquitous in academia. It is a tool mastered by virtually all researchers, offering an intuitive Graphical User Interface (GUI). Choosing the PPTX format ensures *immediate usability*: users can refine colors, layouts, or text without learning new software, minimizing the interaction cost of human-AI collaboration.

2. **Code-Friendliness & Automation Ecosystem.** From a systems engineering perspective, PPTX offers distinct advantages in automated generation. **Structured Standards:** Based on the OpenXML standard, PPTX files are highly structured. Graphical elements (shapes, connectors, text boxes) have clear semantic definitions rather than being mere collections of vector paths. **Python Integration:** PowerPoint is supported by mature Python libraries (e.g., `python-pptx`), ensuring high compatibility with the current AI tech stack. This allows us to model visualization as an *Object-Oriented Code Generation* task—a domain where LLMs excel. In comparison, automating Illustrator relies on proprietary scripting languages (e.g., JSX), hindering seamless integration with deep learning frameworks.

3. **Seamless Workflow Integration.** Research dissemination involves both manuscript publication and conference presentations. **Cross-Scenario Reuse:** Traditionally, researchers must rasterize PDF charts into screenshots for presentations, losing vector quality and editability. **Native Compatibility:** LiveFigure produces native PPTX assets, which are inherently compatible with conference slides and posters. Generated figure objects can be directly copied into presentation decks while maintaining vector resolution and editability, bridging the format gap between manuscript preparation and research presentation.

4. **Decoupling Generation from Refinement.** Given the capabilities of current generative models, we adopt a "Human-AI Collaboration" design philosophy. **Complementary Strengths:** The model handles labor-intensive structure and spatial layout, while the user handles aesthetic judgment and semantic refinement. **Optimal Interface:** PPTX serves as the optimal middleware for this division of labor. The system delivers an editable source file with accurate structure, leaving the final 10% of stylistic polishing to the user via the GUI. This decoupling maximizes AI efficiency while retaining human control over the final visual output.

## B. Supplementary Experimental Results

### B.1. Evaluation of Information Fidelity

To rigorously assess the fidelity with which the generated figures convey the exact scientific information provided in the user's initial input, we designed a dedicated evaluation protocol. Specifically, under this protocol, all models were evaluated using the "Caption + Method Description" (V2) input setting to provide comprehensive initial information. The evaluation

pipeline consists of three stages:

- **Visual Interpretation:** We input the generated illustrations from the test set into a VLM (GPT-4o). The model was tasked with purely visually interpreting the illustrations and generating a detailed description of the content and workflow, strictly without access to any prior contextual prompts.
- **Intent Comparison:** Subsequently, we employ GPT-5 as an evaluator to compare the extracted visual descriptions against the user's original input intent (i.e., the ground-truth Caption and Method Description).
- **Quantitative Assessment:** Based on this comparison, the evaluator model assigned an "Intent Conveyance Score" ranging from 0 to 5. This metric specifically evaluates critical dimensions such as the omission of core methodologies and the correctness of logical relationships between modules.

The quantitative results of this evaluation are summarized in Table 4. The results demonstrate the advantages of the LiveFigure framework in maintaining the information fidelity and interpretability of complex scientific illustrations.

| Models | Average Intent Conveyance Score ↑ |
|---|---|
| Nano Banana | 4.3 |
| GPT-image-1.5 | 3.9 |
| Qwen-image | 2.8 |
| grok-2-image | 1.4 |
| imagen-4.0-ultra | 2.5 |
| Mermaid | 1.2 |
| Graphviz | 3.3 |
| Matplotlib | 2.7 |
| TikZ | 0.8 |
| HTML/CSS | 3.0 |
| **LiveFigure (Ours)** | **4.6** |

*Table 4.* Quantitative comparison of Intent Conveyance Scores across different models.

## B.2. Quantitative Stage-wise Evaluation

While the main manuscript provides a component-wise ablation study, this section presents a progressive, stage-wise evaluation to elucidate the incremental performance gains achieved as the data flows sequentially through our generation pipeline. We sequentially added each stage of the LiveFigure pipeline to a vanilla LLM baseline, tracking Executability, Debug Turns, and the final VLM Score. The quantitative progression is detailed in Table 5.

| Pipeline Configuration | Executability ↑ | Debug Turns ↓ | VLM Score ↑ |
|---|---|---|---|
| Vanilla LLM Generation | 31.0% | 3.40 | 5.03 |
| + Stage 1 (Visual Prior) | 32.5% | 3.50 | 5.75 |
| + Stage 2 (Skills & Experience) | 83.5% | 0.46 | 6.78 |
| + Stage 3 (Refinement) → Full LiveFigure | 83.5% | 0.53 | 7.28 |

*Table 5.* Quantitative progression of the stage-wise evaluation.

By dissecting the trends within these metrics, we can isolate the exact contribution of each module in addressing the dual challenges of execution efficiency and generation quality:

- **Baseline to Stage 1 (Visual Planning Gain):** Introducing the Stage 1 Visual Prior provides the system with human-like macro-layout planning. Consequently, the VLM Score improves from 5.03 to 5.75 due to more logical spatial arrangements. However, Executability remains low (32.5%), and Debug Turns slightly increase (from 3.40 to 3.50).

This authentically reflects that forcing a vanilla LLM to faithfully render a complex visual blueprint using raw, unconstrained Python code actually increases the coding difficulty, leading to more trial-and-error.

- **Stage 1 to Stage 2 (Execution Efficiency Improvement):** Introducing Stage 2 (Skill Library and Debug Experience) completely resolves the aforementioned execution bottleneck. Executability skyrockets from 32.5% to 83.5%, while Debug Turns plummet to 0.46. By encapsulating rendering logic into structured APIs, the system translates abstract layouts into functional code almost in a single pass. This massively improves execution efficiency and boosts the VLM Score to 6.78, as the planned layouts are now accurately and reliably rendered.

- **Stage 2 to Full LiveFigure (Publication Quality Polish):** Stage 3 acts as the visual micro-aligner. In this stage, the agent executes targeted, minor parameter adjustments based on visual feedback to address subtle imperfections in spatial layout, graphic rendering, and element attributes exposed in the initial draft. This highly focused refinement effort significantly elevates the VLM Score to our final state-of-the-art performance of 7.28, quantifying its indispensable role in bridging the gap to publication-ready quality.

### B.3. Case Study 1: Cross-Model Comparison of Outputs Given Identical Inputs

In this case, the input comes from the caption and method description of the ICLR 2024 paper "Unsupervised Order Learning" (UOL) (Lee et al., 2024). As shown in Figure 6, the visualization of the UOL algorithm requires depicting a complex nested logic: an outer *Alternating Optimization* loop containing two distinct sub-modules (*Ordered k-means* and *Network Fine-tuning*). LiveFigure (Ours) successfully captures this hierarchical logic. It organizes the sub-modules within a clear container structure and visualizes the sequential dependency ($\mu_1 \to \cdots \to \mu_k$) with appropriate spatial arrangement. Crucially, LiveFigure maintains a publication-friendly aspect ratio (e.g., 4:3 or 16:9), ensuring the diagram is legible and compact. In contrast, baseline models exhibit significant limitations regarding layout control and aspect ratio adaptability:

Code-based Tools (TikZ, Graphviz, Mermaid): These tools rely on rule-based rendering engines that often ignore the semantic importance of canvas dimensions. For instance, the TikZ baseline (bottom right) generates an extremely wide, strip-like image. This is a structural limitation of LLM-generated code using the `\documentclass[tikz]{standalone}` class; the compiler creates a content-aware crop that strictly wraps the generated nodes. Without explicit, complex layout constraints from the LLM, the output dimensions are dictated purely by the natural flow of elements, often resulting in unusable aspect ratios (e.g., extremely vertical or horizontal layouts) that require significant post-processing to fit into a paper.

Raster Generative Models (e.g., Qwen-image, Imagen): While aesthetically pleasing, these models are often constrained by API specifications to fixed aspect ratios (e.g., 1:1 squares). This limitation forces the model to cram complex, linear workflows into a square canvas, leading to cluttered layouts or illogical "snaking" paths that disrupt the visual flow of the algorithm (as seen in the Qwen-image and grok-2 examples).

LiveFigure effectively bridges this gap by decoupling logical generation from layout rendering, ensuring both structural fidelity and adaptable, professional formatting.

### B.4. Case Study 2: Cross-Model Comparison of Outputs Given Identical Inputs

In this case, the input comes from the caption and method description of the ICLR 2024 paper "Decoding Natural Images from EEG for Object Recognition" (Song et al., 2023). As illustrated in Figure 7, this task presents a higher-order topological challenge. The figure requires a composite layout consisting of two distinct sections, a high-level training/inference framework (Panel A) and a fine-grained neural network architecture (Panel B). LiveFigure (Ours) successfully disentangles this complex logic by autonomously identifying the need for a bipartite layout. It generates two distinct, clear sub-figures that respect the semantic hierarchy. Panel A cleanly delineates the contrastive learning setup separating the training and testing phases, while Panel B accurately reconstructs the internal data flow of the encoder, sequentially arranging specific technical modules such as Temporal-Spatial Convolution and Graph Attention. The result is a legible, publication-ready schematic that perfectly balances high-level abstraction with low-level technical specification.

In stark contrast, raster-based generative models (e.g., Qwen-image, Grok-2) struggle to maintain this structural separation. Constrained by fixed aspect ratios (typically 1:1) and a lack of explicit layout planning, these models often conflate the two distinct panels into a single, cluttered scene. Instead of producing a rigorous technical diagram, models like Grok-2 and Qwen-image tend to generate artistic, "sci-fi" style illustrations, such as glowing brain renderings, that prioritize visual impact over methodological fidelity. Consequently, they frequently hallucinate non-existent connections or omit critical internal modules, for example, the specific attention blocks in Panel B, rendering the output aesthetically striking but

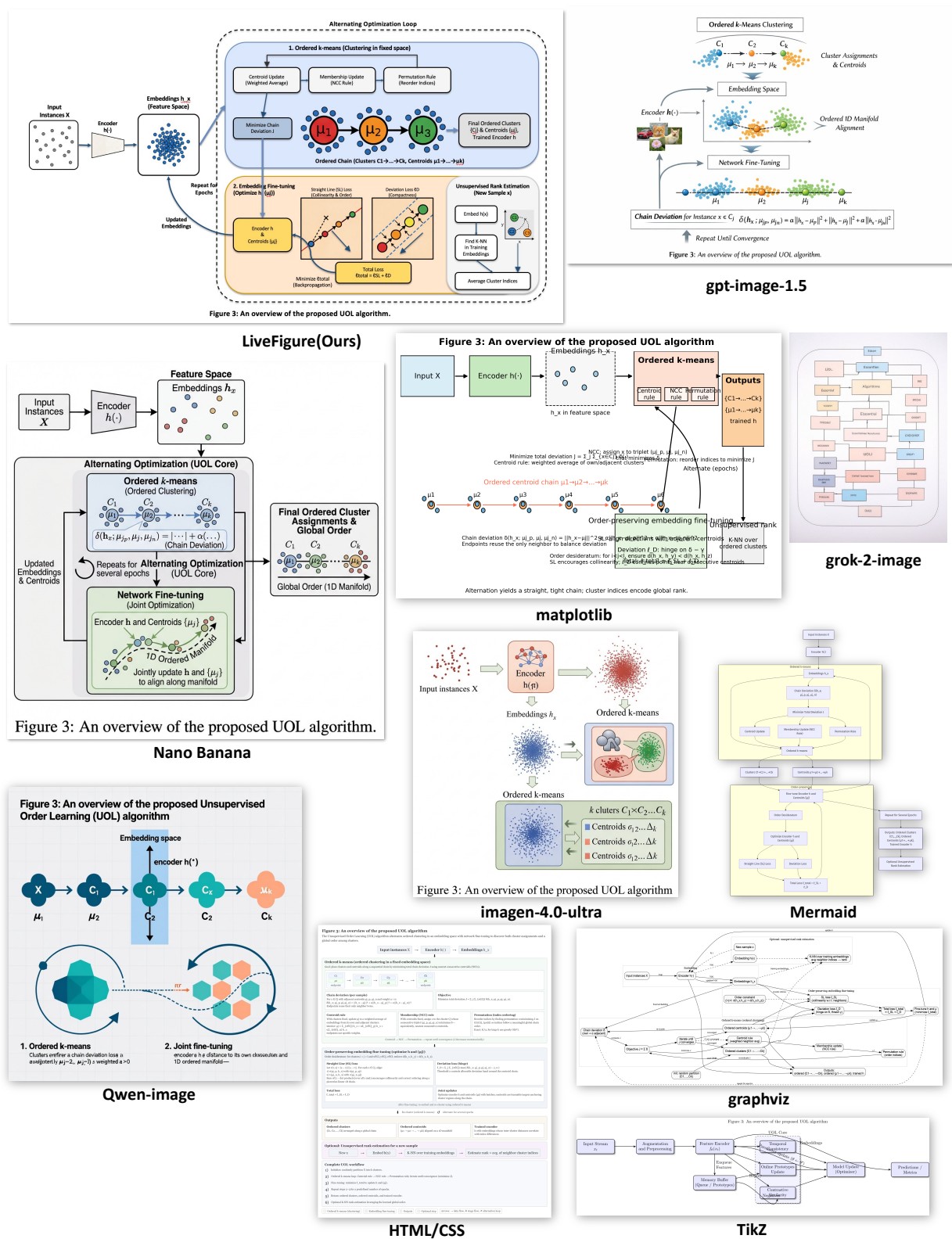

*Figure 6.* Qualitative comparison of generated figures for the *Unsupervised Order Learning (UOL)* paper. We compare LiveFigure against state-of-the-art raster generative models and code-based plotting tools.

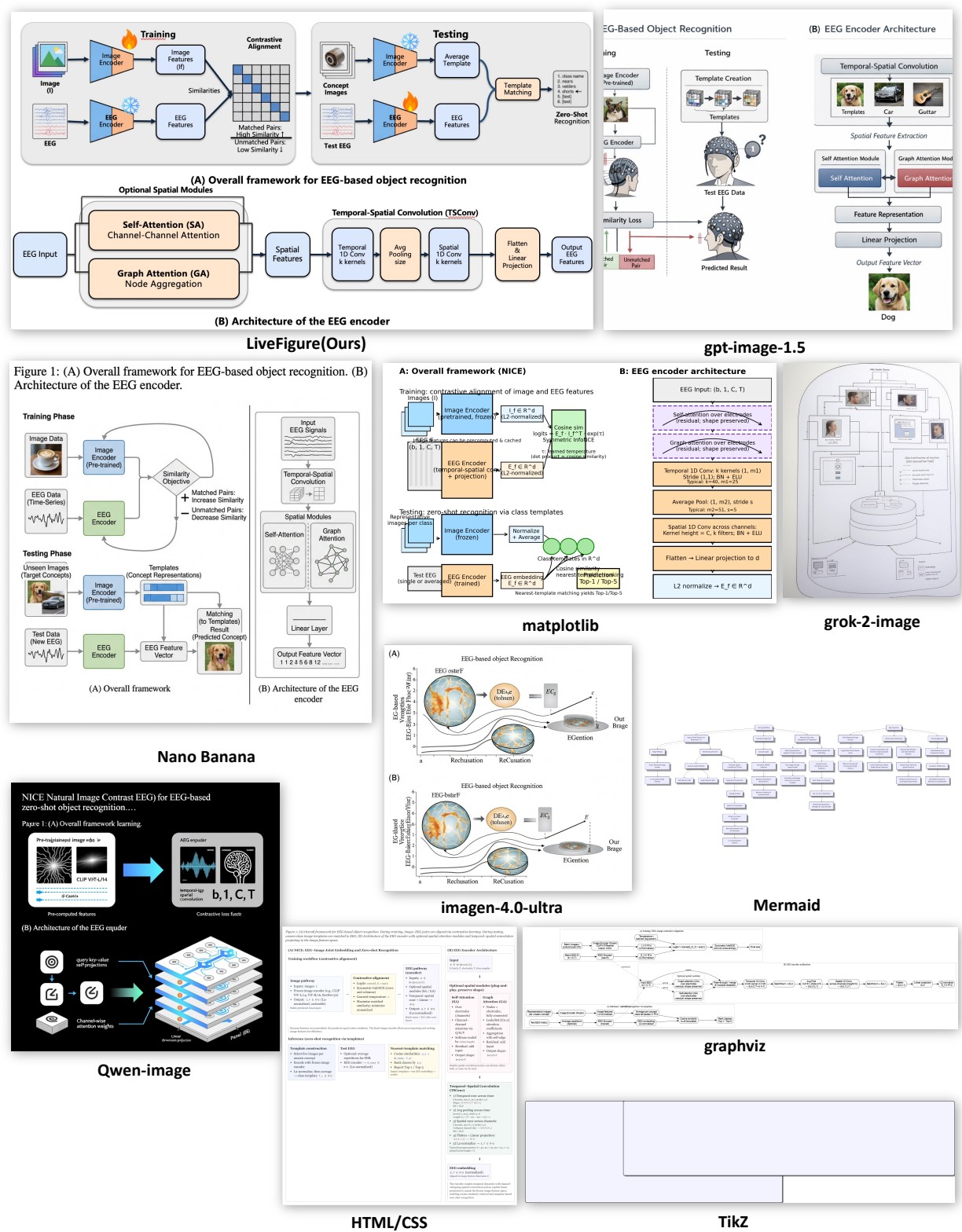

*Figure 7.* Qualitative comparison of generated figures for the *Decoding Natural Images from EEG for Object Recognition* paper.

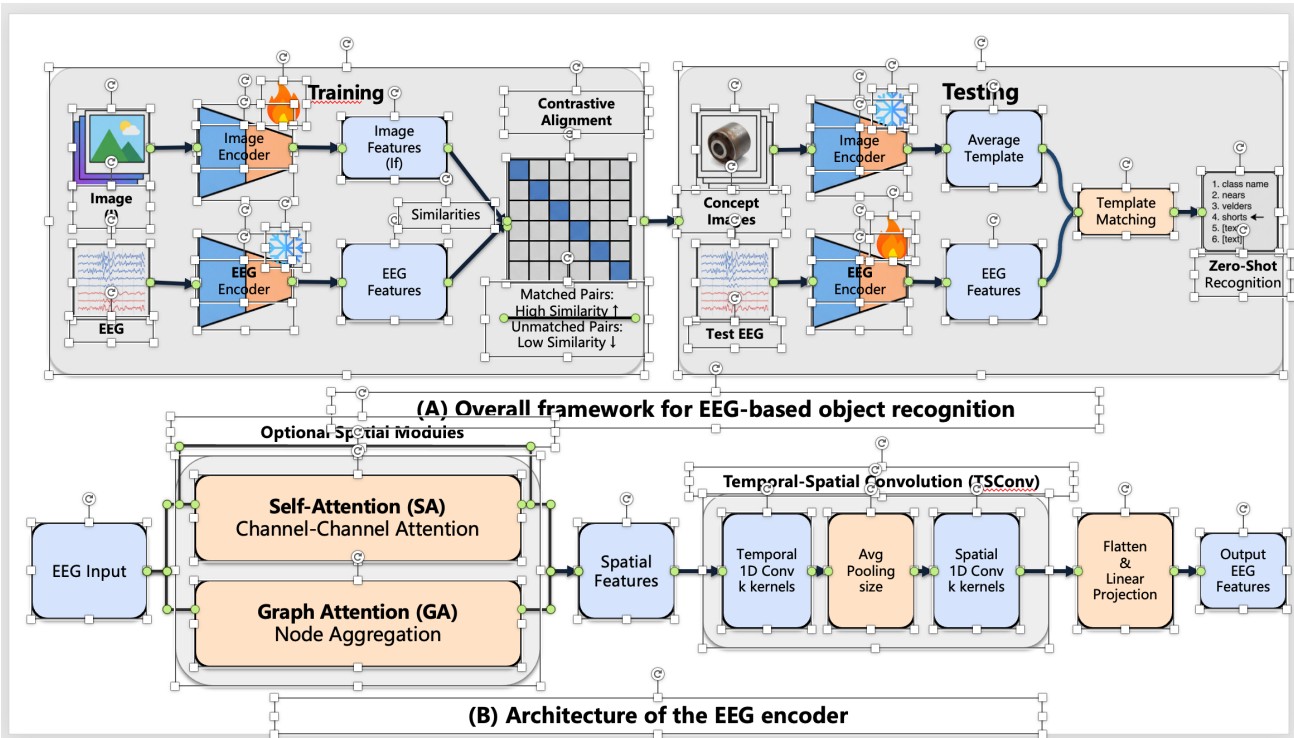

*Figure 8.* Visual verification of object-level editability. This screenshot captures the generated figure within the Microsoft PowerPoint interface with all elements selected. The visible bounding boxes and control handles confirm that the output consists of discrete, manipulatable native objects (shapes, text boxes, connectors) rather than a flattened raster image.

technically inaccurate.

Similarly, code-based baselines such as Matplotlib, Mermaid, and Graphviz exhibit severe rigidity when handling such multi-part logic. Lacking a semantic understanding of sub-figures, these tools typically force the entire workflow into a single linear flowchart, effectively erasing the distinction between the global framework and the local architecture. The Matplotlib baseline, in particular, illustrates a catastrophic failure mode common in standard plotting libraries. Lacking native primitives for schematic drawing, it fails to render the architecture entirely, outputting meaningless coordinate axes. By effectively decoupling logical planning from asset generation, LiveFigure bridges these gaps, producing a coherent multi-panel figure that traditional tools and end-to-end models fail to construct.

### B.5. Case Study 3: Verification of Full Editability and Object Granularity

We present the third case to explicitly verify the granular editability and object-oriented nature of our outputs. Unlike pixel-based generative models that produce flattened raster images where elements are fixed, LiveFigure generates native, manipulatable source files. Figure 8 displays a raw screenshot of the generated EEG framework opened directly within the Microsoft PowerPoint interface, with a global selection command applied to reveal the underlying structure. The dense array of visible selection handles and bounding boxes serves as rigorous visual proof that the output is composed of discrete, native graphical primitives—including shapes, text boxes, and smart connectors—rather than a static pixel matrix.

This object-level granularity grants researchers full control over every visual element. As evidenced by the specific text resizing handles, every label is instantiated as a standalone text box, allowing users to directly correct typos, adjust font properties, or rewrite content without the need for complex image inpainting techniques. Furthermore, the connecting arrows are generated as dynamic PowerPoint connectors attached to shape anchors; this means that relocating a module (e.g., dragging the "Spatial Features" block) automatically triggers a rerouting of the associated arrows, preserving the topological logic of the diagram. Consequently, the entire figure is vector-based and resolution-independent, enabling it to be resized or restyled indefinitely to fit various presentation contexts, effectively bridging the gap between AI generation and professional design workflows.

## B.6. Case Study 4 in Biology: Cross-Disciplinary Generalization

To demonstrate the cross-disciplinary generalizability of the LiveFigure framework, we provide an additional case study from the domain of biology. Figure 9 illustrates the separation and purification workflow of Extracellular Vesicles (EV). Specifically, the rendered schematic in Figure 9(a) depicts the classic combination of differential centrifugation and density gradient centrifugation, widely utilized for extracting exosomes. It demonstrates how LiveFigure successfully models specialized biological concepts, multi-step experimental processes, and domain-specific visual elements. Additionally, Figure 9(b) displays the native PowerPoint UI, confirming that the output retains full editability. This case study highlights the adaptability of our agentic orchestration and demonstrates that LiveFigure's procedural generation paradigm generalizes effectively beyond standard computer science and AI system architectures.

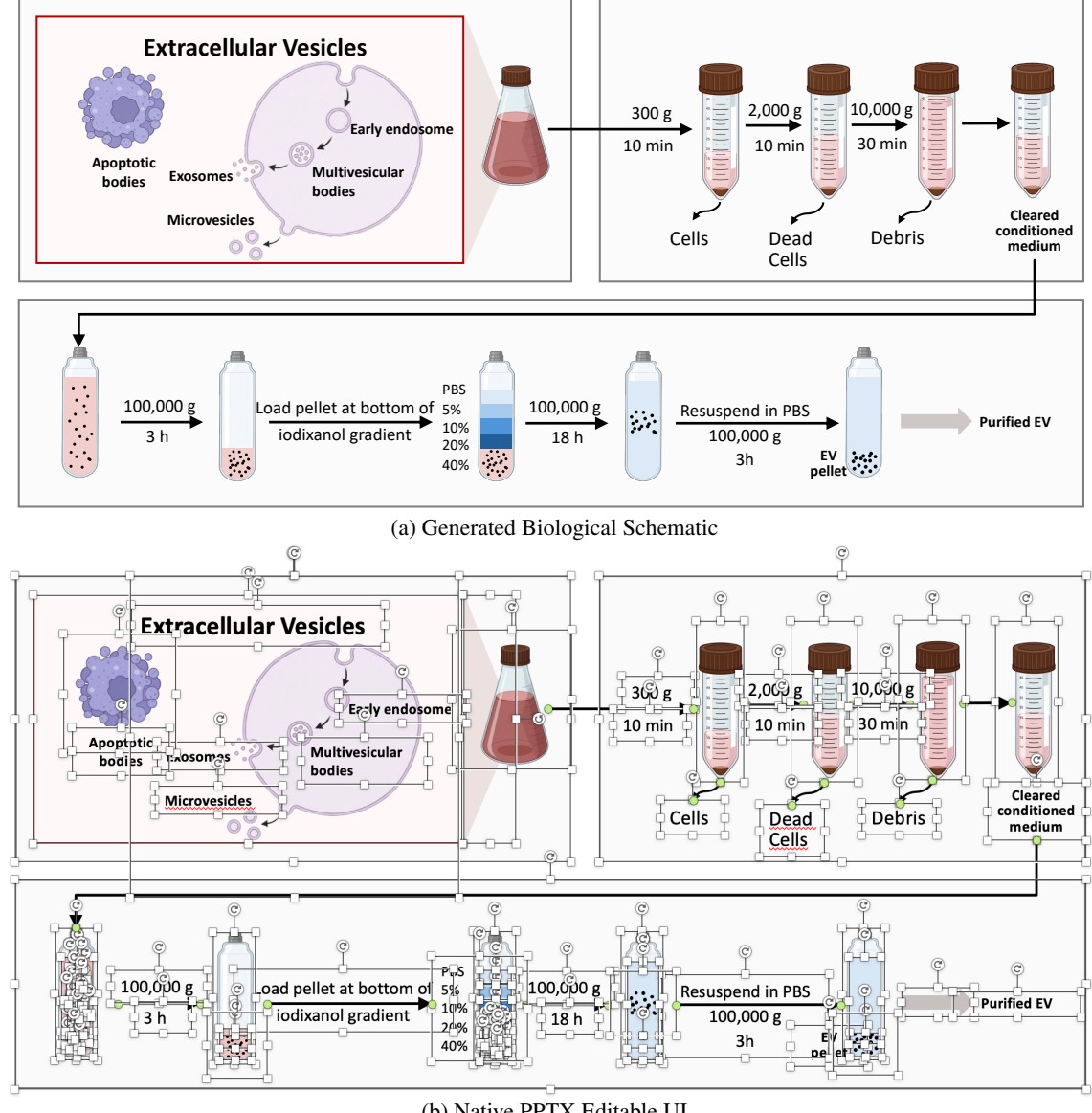

(a) Generated Biological Schematic

(b) Native PPTX Editable UI

*Figure 9.* LiveFigure generation for a biological workflow. (a) The rendered schematic. (b) The natively editable output in PowerPoint.

## B.7. Case Study 5: Failure Case Analysis

While the LiveFigure framework demonstrates robust performance and stability in the majority of scenarios, it occasionally encounters limitations when processing complex system architectures with exceptionally high information density. Specifi-

cally, in diagrams with a massive number of interconnected modules, the system may produce overly crowded layouts or "spaghetti routing" issues.

To illustrate this limitation, we provide a failure case analysis in Figure 10. This generated schematic contains approximately 44 textual nodes and nearly 30 logical connections. Due to the extreme local node density and constrained spatial layout, the underlying obstacle-avoidance routing algorithm occasionally fails. Consequently, certain connection lines, such as the arrow linking the "MoV-Adapter" to the "Large Language Model," fail to intelligently detour around obstacles. Instead, the line directly penetrates the physical boundary of a text box, resulting in a visual artifact.

However, a significant mitigation to this algorithmic limitation lies in the natively editable format of LiveFigure's outputs. Unlike rasterized images generated by traditional models, where such an error would necessitate regenerating the entire figure, the generated PPTX file allows researchers to manually correct this routing failure effortlessly. By simply dragging and dropping the anchor points of the affected connector within PowerPoint, the visual artifact can be perfectly resolved in a matter of seconds.

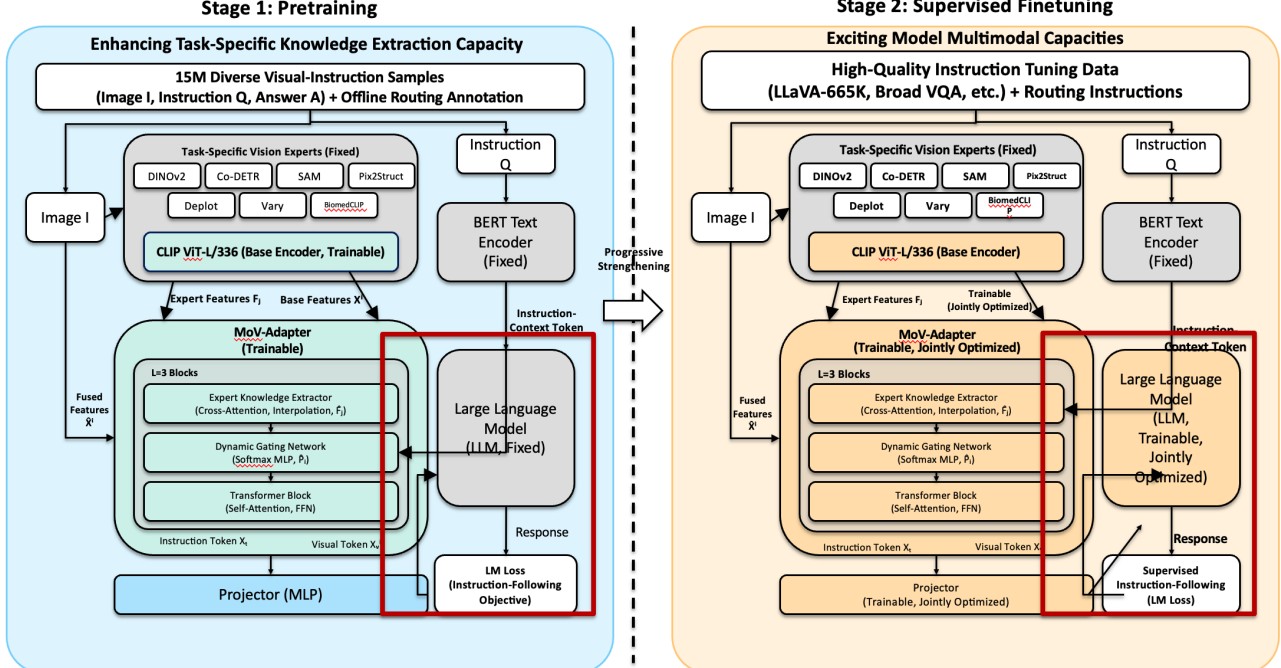

Figure 3: The training strategy of MoVA. We enhance the task-specific knowledge extraction capacity in the first stage. Then, we excite model multimodal capacities in the last stage.

*Figure 10.* A failure case exhibiting "spaghetti routing" in a highly dense system architecture diagram. Due to severe spatial constraints, certain connection lines (e.g., between the "MoV-Adapter" and the "Large Language Model") overlap with textual boundaries. Thanks to the native editability of the output, such artifacts can be manually fixed in seconds by adjusting the anchor points.

### B.8. Case Study 6: Natural Language-Based Interactive Modification

A key advantage of the LiveFigure framework is its native support for natural language-based interactive modification. Benefiting from the code-driven procedural generation paradigm, these modifications are highly precise and efficient. Unlike raster-based image generation models that require full-image regeneration for minor tweaks, the elements in our generated figures correspond to independent, structured code segments.

When a user provides a modification instruction (e.g., "Change the color of the Encoder module to blue"), the Coding Agent precisely locates and modifies only the targeted code snippets. After verification by the Critique Agent, the updated code is executed to export the revised vector graphic. This mechanism ensures that the original layout and topological structures remain strictly intact during the editing process. To demonstrate this capability, we present a case study illustrating consecutive interactive modifications in Figure 11:

- **Initial Generation:** Figure 11(a) displays the originally generated flowchart.

- **Edit Instruction 1:** "Recolor the rounded rectangles in the bottom-left 'closed-loop systems' block to light green." As shown in Figure 11(b), the system successfully applies the color change exclusively to the targeted modules without affecting the surrounding elements.
- **Edit Instruction 2:** "Delete the bottom title text box and shift the 'tree search-based methods' block inward, as it currently exceeds the gray background." As shown in Figure 11(c), the system accurately removes the specified text and spatially realigns the target block.

### B.9. Limitations and Future Work

While the LiveFigure framework significantly automates the generation of scientific illustrations, we identify two primary limitations that present promising avenues for future research.

**A Tendency Toward Visual Over-complication.** We observe that large language models exhibit a tendency to over-complicate visual representations when translating textual descriptions into figures. When provided with highly detailed methodological descriptions, the agent frequently attempts to exhaustively visualize every variable and intermediate operational step. Consequently, the output often resembles an exhaustive technical schematic rather than an effective scientific illustration. In practice, researchers prefer to strategically highlight core design innovations and primary contributions while abstracting away secondary details. This mismatch reduces the clarity and communicative effectiveness of the generated figures. To address this issue, we plan to explore alignment-based training methods, such as Reinforcement Learning from Human Feedback (RLHF). By leveraging human preference data, we aim to explicitly train the model in *strategic abstraction* and *hierarchical emphasis*, enabling it to intelligently filter information and highlight key methodological contributions.

**Limitations in Granular Stylistic Control.** LiveFigure can leverage different natural language conditioning to guide the aesthetic style of the generated figures. However, this approach often lacks the precision required for highly customized visual outputs. Researchers often have distinct stylistic preferences, with some favoring formal and disciplined visual designs, while others prefer more engaging and expressive visual styles. Furthermore, different premier venues exhibit distinct visual conventions. For example, figures in *NeurIPS* papers typically favor schematic, modular, and function-oriented designs, with an emphasis on clarity, block-diagram structures, and concise labeling. In contrast, figures in *Nature* publications often adopt a more polished and visually refined style, characterized by balanced composition, consistent typography, subtle color palettes, and a higher degree of visual standardization suitable for broad scientific communication. To achieve more effective personalized preference guidance, our future work involves curating a fine-grained, venue-annotated dataset of scientific illustrations. This dataset will facilitate targeted fine-tuning and the development of stylistic adapters, allowing the framework to precisely and automatically align with specific author preferences or stringent journal requirements.

## C. Details of the Experimental Setup

### C.1. Baseline Models

We compare our approach with several representative image generation systems and widely-used programmatic figure creation libraries. These baselines cover commercial-grade, open-model, and code-driven figure generation methods.

- **Gemini-3-Pro-Image (Nano Banana)**: Released by Google in November 2025 as part of the Gemini 3 series, Gemini-3-Pro-Image is a commercial image generation and editing model that integrates tightly with Google's Gemini ecosystem. It supports high-resolution image generation, multimodal reasoning and contextual consistency, and enhanced text rendering in complex scenes. The model has been widely adopted for general creative use and remains a key competitor in the commercial image generation landscape.
- **GPT-Image-1.5**: Introduced by OpenAI in December 2025, GPT-Image-1.5 is the latest flagship image generation model in the GPT Image family. Designed for production-quality visuals, it significantly improves instruction following, detail fidelity, and generation speed over earlier versions, while reducing generation costs. GPT-Image-1.5 also supports iterative image editing workflows, enabling users to refine outputs while preserving composition and key visual features.
- **Imagen-4.0-Ultra**: Part of Google's latest Imagen 4 family of models, Imagen-4.0-Ultra (officially "imagen-4.0-ultra-generate-001") offers photorealistic image generation with detailed prompt adherence and flexible support for multiple aspect ratios and high resolutions up to 2048×2048. It supports digital watermarking, safety settings, and prompt enhancement, making it suited for professional design, marketing content, and high-fidelity

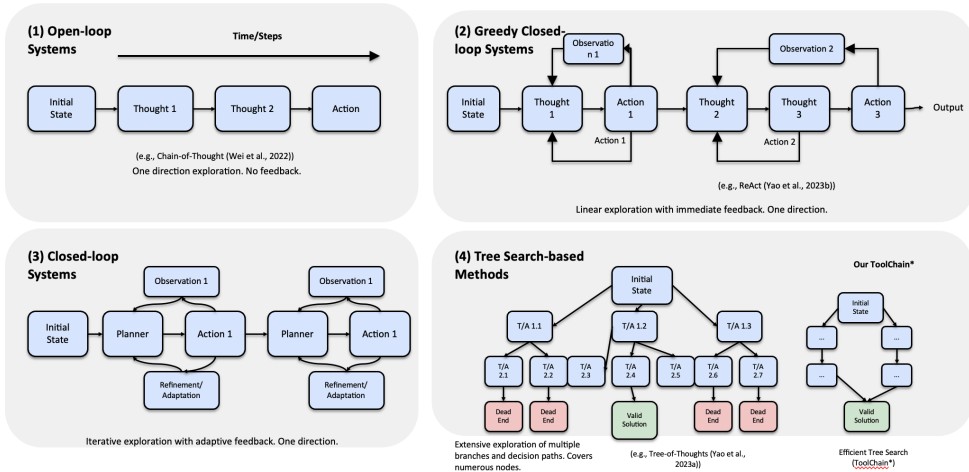

Figure 1: A comparison of existing methods that leverage LLMs for decision-making from a searching space perspective. Most existing methods of (1) open-loop systems (e.g., Chain-of-Thought (Wei et al., 2022)), (2) greedy closed-loop systems (e.g., ReAct (Yao et al., 2023b)), and (3) closed-loop systems (e.g., Adaplander (Sun et al., 2023)) only explore one possible direction. This often leads to limited exploration of the entire action space. In contrast, (4) tree search-based methods (e.g., Tree-of-Thoughts (Yao et al., 2023a)) identify a valid solution path by extensively examining multiple decision space branches, covering almost every conceivable node. Our proposed ToolChain* belongs to the tree search-based category and improves by developing an efficient search algorithm.

(a) Original Generated Figure

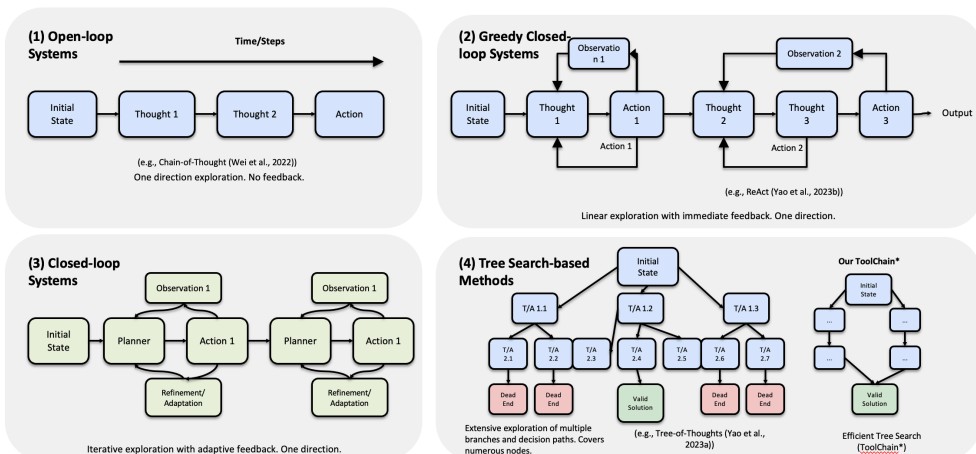

Figure 1: A comparison of existing methods that leverage LLMs for decision-making from a searching space perspective. Most existing methods of (1) open-loop systems (e.g., Chain-of-Thought (Wei et al., 2022)), (2) greedy closed-loop systems (e.g., ReAct (Yao et al., 2023b)), and (3) closed-loop systems (e.g., Adaplander (Sun et al., 2023)) only explore one possible direction. This often leads to limited exploration of the entire action space. In contrast, (4) tree search-based methods (e.g., Tree-of-Thoughts (Yao et al., 2023a)) identify a valid solution path by extensively examining multiple decision space branches, covering almost every conceivable node. Our proposed ToolChain* belongs to the tree search-based category and improves by developing an efficient search algorithm.

(b) After Instruction 1: Recolored modules in the "closed-loop systems" block.

(c) After Instruction 2: Deleted bottom title and realigned the "tree search" block.

*Figure 11.* A sequential case study of natural language-based interactive modification. Thanks to the code-driven paradigm, LiveFigure can accurately modify specific attributes and spatial layouts based on consecutive user instructions while preserving the global topology.

creative tasks.

- **Qwen-Image**: Qwen-Image is an open-model image generation system released by Alibaba in August 2025 as part of the Qwen multimodal model family. Trained on a large-scale diverse dataset with a progressive curriculum learning strategy, Qwen-Image excels at complex text rendering, multilingual handling, and consistent image editing. Its architecture enables robust performance across both alphabetic and logographic languages, making it a strong open-model baseline.

- **Grok-2-Image**: Grok-2-Image is the text-to-image generation variant of xAI's Grok-2 model, available via the xAI image API. It generates sharp and photorealistic visuals from natural-language prompts and is optimized for fast, API-driven workflows. While its open access and flexible usage make it popular for prototype and creative content generation, it has also faced regulatory and safety discussions due to misuse in generating inappropriate content.

In addition to these image generation models, we also consider widely-used programmatic libraries for figure creation:

- **Mermaid**: A declarative code-based diagramming library that generates flowcharts, sequence diagrams, class diagrams, and Gantt charts from simple textual descriptions.
- **Graphviz**: An open-source graph visualization software that uses the DOT language to render structured diagrams such as flowcharts, dependency graphs, and network visualizations.
- **Matplotlib**: A widely adopted Python plotting library that supports a diverse range of 2D charts and figure customization options, commonly used in scientific and engineering visualization workflows.
- **TikZ-V1**: A LaTeX-based vector graphics package for producing publication-quality diagrams and figures directly within LaTeX source files, offering precise control over layout and style.
- **HTML/CSS**: Standard web technologies used to programmatically construct and style visual figures, charts, and layouts in document or web-rendered formats.

For the experimental comparisons, the inputs provided to these code-based baselines are strictly identical to those used for LiveFigure, utilizing either the Caption-only (V1) or Caption + Method Description (V2) settings. All code-based baselines are implemented using the Gemini-3-Pro. Specifically, we directly prompt the Gemini-3-Pro to generate the complete executable code (e.g., DOT language for Graphviz or a Python script for Matplotlib) based on the given text descriptions. The generated code is then automatically compiled and executed within a local sandbox environment to render and export the final PNG or PDF images. Finally, these outputs are evaluated using a protocol that is completely consistent with the evaluation of LiveFigure.

### C.2. Interface for Human Preference Voting

To evaluate the quality of the generated scientific figures, we conducted a blind A/B preference test with domain experts (Section 4.5) using a custom Python-based graphical user interface (GUI). The evaluation interface was specifically designed to facilitate unbiased comparison by presenting two figures side-by-side: one generated by our proposed method ("Ours") and the other by a baseline method (e.g., Matplotlib, TikZ, or another model). Crucially, the evaluation is completely blind; the position of the figures (left vs. right) is randomized for each trial, and all identifying labels such as model names are hidden from the evaluator, with options labeled simply as "Image Left" and "Image Right".

Evaluators interact with the system by choosing "Left is Better", "Right is Better", or "Tie / Skip", with keyboard shortcuts implemented for efficiency. The visual presentation is optimized for clarity, with images automatically scaled to fit the display area while maintaining their aspect ratio, ensuring details are visible regardless of original resolution, all within a clean, minimalist "Academic Blue & Grey" color theme to minimize distraction. The process begins with the automatic generation of image pairs for each paper in our test set, retrieving the figure from our method and a corresponding one from a randomly selected baseline, shuffled to ensure random presentation order. Expert reviewers, familiar with scientific data visualization, are instructed to judge based on visual quality, clarity, and adherence to plotting standards. All decisions are logged automatically, capturing timestamp, paper ID, the winner, and the specific models involved, providing the raw data for the win rates reported in Section 4.5. This rigorous framework ensures that our comparative results reflect genuine human preference, mitigating potential biases related to method recognition or presentation order.

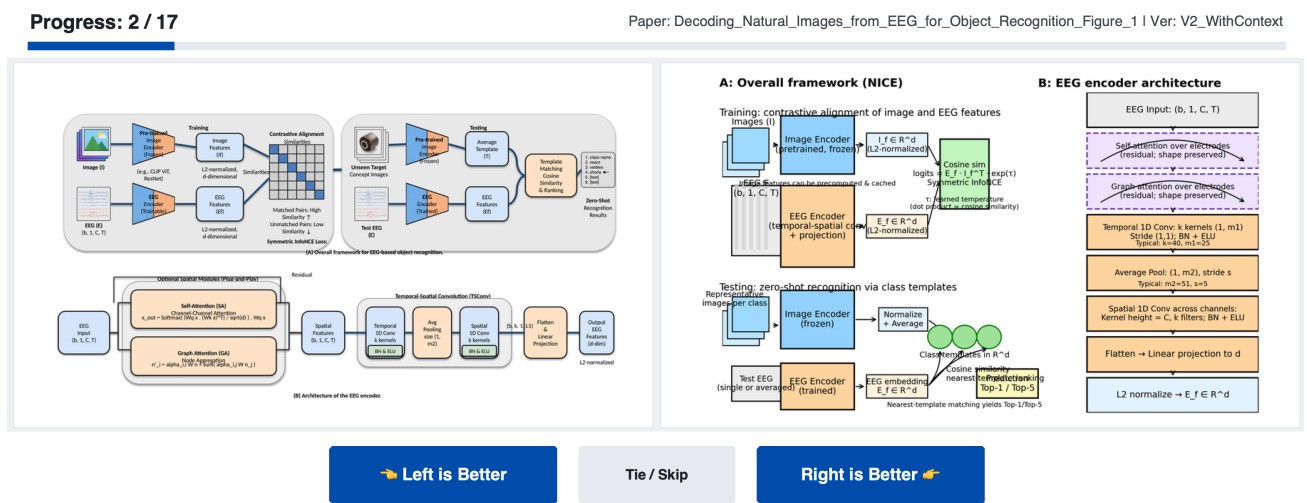

*Figure 12.* The interface for human preference voting. Participants are not exposed to the identities of the underlying models associated with the images.

## C.3. Prompts for Key Components

Experiences distilled from past erroneous coding records and debugging sessions are formulated into prompts that are incorporated into future coding inputs. Due to space limitations, only a subset of the experiences is shown here. Please refer to our code repository for the complete set of documented experiences.

```
PPTX_BEST_PRACTICES = """
*** CRITICAL PYTHON-PPTX RULES (MUST FOLLOW) ***
1. **Lines are CONNECTORS**:
   - NEVER use `slide.shapes.add_shape(MSO_SHAPE.LINE, ...)` -> This causes
     AttributeError.
   - ALWAYS use `slide.shapes.add_connector(MSO_CONNECTOR.X, ...)`.
   - **Valid Types**: `MSO_CONNECTOR.STRAIGHT`, `MSO_CONNECTOR.ELBOW`, `
     MSO_CONNECTOR.CURVE`.
   - **INVALID**: Do NOT use `MSO_CONNECTOR.CURVED` (No 'D' at the end).
   - **INVALID SHAPES**: `MSO_SHAPE.DOC_TAG` does not exist (Use `MSO_SHAPE.
     FOLDED_CORNER` instead).

2. **Connector Properties**:
   - Connectors (Lines/Arrows) have `.line` but **NO `.fill`**.
   - NEVER try to set `connector.fill.solid()`. Only set `connector.line.color.rgb`.

3. **Shape Fills (NO ONE-LINERS)**:
   - **NEVER** try to create and color a shape in one line: `add_shape(...).fill.
     fore_color.rgb = ...` (This crashes with TypeError).
   - **ALWAYS** split into steps:
     1. `shape = slide.shapes.add_shape(...)`
     2. `shape.fill.solid()`  <-- REQUIRED first!
     3. `shape.fill.fore_color.rgb = RGBColor(...)`
"""
```

We created a documentation for the predefined and debugged plotting skills, detailing each skill's functionality, invocation method, parameter choices, and other relevant aspects. Some of the skills are described as follows. Due to space limitations, the prompts shown here only include the first skill as a representative example. Please refer to our code repository for the complete documentation of all skills.

```
SKILLS_SPECIFICATION = """
*** HIGH-PRIORITY SKILLS SPECIFICATION ***

You have access to a local library 'skills.py' that provides high-level plotting
    capabilities.
**You must always prioritize using these skills over native 'python-pptx' methods to
     ensure best visual quality.**

---

#### **Global Rules & Best Practices**

1. **Imports**: **ALWAYS** use wildcard import to get all skills:
'''python
    from skills import *
'''

2. **Coordinate Units**:
* For **Skills** (e.g., 'add_block', 'add_connector'): Use **raw floats** (e.g., '
    left=5.0').
* For **Native PPTX** ('slide.shapes.add_shape'): Use **'Inches()'** (e.g., 'left=
    Inches(5.0)').

3. **Routing**: Do not calculate connection indices manually. The skills handle
    alignment automatically.
4. **Objects**: Always pass Shape/Picture objects to connector functions ('
    add_connector'), not their names.
5. **Strict Parameter Compliance**: The function signatures listed below are
    EXHAUSTIVE. DO NOT use any parameters that are not explicitly defined in the
    documentation (e.g., do not hallucinate linestyle, dashed, shadow, or end_arrow
    unless they appear in the signature).

---

### **SECTION 1: UNIVERSAL DRAWING SKILLS (Nodes, Text, Groups)**

#### **Skill 1: 'add_container' (Background Grouping)**

**Description**
Draws a background rectangle to visually group related elements.

* **Best Practice**: Call this **FIRST** (Layer 1) before drawing nodes inside it,
    to ensure it stays in the background.

**Function Signature**

'''python
add_container(slide, x, y, w, h, title=None, fill_color='F5F5F5', stroke_color='
    CCCCCC', alpha=1.0)

'''

**Parameters**

* **'title'** *(str, optional)*: Automatically adds a bold title at the top inside
    the container.
* **'alpha'** *(float)*: Transparency. '1.0' is opaque, '0.0' is invisible. Use
    '0.1'-'0.3' for subtle backgrounds.

**Example**
```

```python
'''python
# Draw a light grey background area for the "Encoder" section
group_box = add_container(slide, x=0.5, y=1.0, w=4.0, h=5.0, title="Encoder Layers")

'''
"""
```

The prompts for Module Assembly are as follows, where Python code is generated based on the skills documentation and experience summarized from past debugging processes.

```
BLUEPRINT_TO_CODE_PROMPT = """
You are an expert Python developer specialized in 'python-pptx'.

Task:
Write a COMPLETE, STANDALONE Python script to reconstruct a scientific diagram from
    the given image.

Context Requirements:
1. Objective:
    Create a scientific diagram based on the user's request:
    "{requirement}"

2. Layout Reference:
    - Mimic the attached image's overall structure, spatial layout, shapes, arrows,
        and text.
    - Preserve relative positioning and visual hierarchy as closely as possible.

3. Text Guidelines:
    - Always use BLACK as the text color.
    - All text inside shapes or text boxes MUST be center-aligned.
    - Font size should be clearly readable and proportionate to the corresponding
        shapes.
    - Avoid excessively small text.

4. Coordinate Precision:
    - Pay close attention to the precise placement of all shapes and text.
    - Coordinates directly determine alignment and the overall visual quality of the
        figure.
    - Sloppy alignment is unacceptable.

Technical Specifications:
1. Canvas Size:
    - Width = {w_cm} cm
    - Height = {h_cm} cm
    - You may adjust the canvas size ONLY if absolutely necessary.

2. Output:
    - You MUST save the presentation EXACTLY as "{output_filename}".

3. Imports:
    - Include ALL required imports explicitly.
    - This includes (but is not limited to):
        Presentation, Cm, Inches, RGBColor, MSO_AUTO_SHAPE_TYPE, PP_ALIGN, etc.

{asset_prompt_section}

Best Practices:
{PPTX_BEST_PRACTICES}
```

```
Tooling and API Constraints:
{TOOLS_SPECIFICATION}

IMPORTANT OUTPUT FORMAT (STRICT):
1. Output RAW Python code ONLY.
2. DO NOT use Markdown code blocks (no ```python).
3. DO NOT include any explanations, comments outside code, or natural language text.
4. The output MUST:
   - Start directly with import statements
   - End with the presentation save command
"""
```

When a bug occurs, the prompts for debugging are as follows:

```
DEBUG_CODE_PROMPT = """
The following Python script failed to execute.

--------------------------------------------------
[Error Log]
{error_log}
--------------------------------------------------

--------------------------------------------------
[Broken Code]
{broken_code}
--------------------------------------------------

Task:
1. Analyze the Error Log to identify the syntax or logical issue.
2. Fix the code to resolve the error.
3. Ensure the code saves the output EXACTLY as "{output_filename}".
4. Return the COMPLETE and FIXED Python script.
5. For parts of the code that do not involve errors, DO NOT modify them.

Best Practices:
{PPTX_BEST_PRACTICES}

Tooling and API Constraints:
{TOOLS_SPECIFICATION}

IMPORTANT OUTPUT FORMAT (STRICT):
1. Output RAW Python code ONLY.
2. DO NOT use Markdown code blocks (no ```python).
3. DO NOT explain the fix or include any natural language text.
4. The output MUST start directly with import statements.
"""
```

The input prompts for a VLM that acts as a "visual critic" to perform diagnosis and output a structured Actionable Issue List are as follows.

```
CRITIQUE_VISUAL_PROMPT = """
You are a Senior Design QA Engineer for scientific publications.

Role & Goal:
- You are given a single image representing the Current Result of a scientific
   figure.
```

- Your goal is to diagnose VISUAL, STRUCTURAL, and PRESENTATIONAL deficiencies in
    the current figure
  and provide precise, actionable recommendations to improve its clarity,
      correctness, and
  publication-quality appearance.

Task:
Perform a STRUCTURED VISUAL INSPECTION of the Current Result by strictly following
    the checklist below
and produce EXECUTABLE, ELEMENT-BOUND FIX SUGGESTIONS.

CRITICAL REQUIREMENT: BE SPECIFIC
- BAD: "Fix the arrow."
- GOOD: "Change the arrow connecting 'Input' and 'Model' to an Elbow Connector."
- GOOD: "The arrow head is too large; reduce it to Medium size."

--------------------------------------------------

INSPECTION CHECKLIST (Evaluate ALL 4 Dimensions):

1. CANVAS & BOUNDARIES (CRITICAL)
    - Check whether any content (especially near the RIGHT or BOTTOM edges) is
        clipped or cut off.
    - Common failures: shapes, labels, or arrows exceeding slide boundaries.
    - Fix Advice Examples:
      - "Move [Specific Element Name] LEFT/UP to avoid clipping"
      - "Shift ALL elements LEFT by a small margin"
    - If absolutely necessary, adjusting the canvas size is allowed.

2. CONNECTOR LOGIC & STYLE (CRITICAL)
    - Check whether any arrows cross THROUGH text boxes or shapes instead of routing
        around them (SEVERE ERROR).
    - Check whether arrow start/end points attach to the correct side of nodes.
    - Style Checks:
      - Arrowhead size (too large or clumsy?)
      - Line width (too thick like a stick or too thin to see?)
      - Scientific figures typically prefer:
        - Line width: 1.5 pt     2.0 pt
        - Arrowhead size: Medium or Small
    - Fix Advice Examples:
      - "Reroute the arrow between [A] and [B] to avoid crossing [C]"
      - "Change connector type to Elbow"
      - "Reduce arrowhead size to Medium"
      - "Set line width to 1.5 pt"

3. TEXT INTEGRITY
    - Check whether text spills out of its container.
    - Check whether font size is too large (crowded) or too small (unreadable).
    - Check font color:
      - Text should be BLACK or dark gray.
    - Fix Advice Examples:
      - "Move [Specific Text Box] RIGHT by approximately [distance]"
      - "Change [Specific Label] font color to BLACK"
      - "Widen [Specific Shape] to fit the text"

4. VISUAL ALIGNMENT & STYLE
    - Check whether the logical layout structure matches the Reference Goal.
    - Check whether colors look professional and publication-ready.
    - Avoid neon or overly light colors unless they are semantically required.

--------------------------------------------------

OUTPUT REQUIREMENTS (STRICT):

```
- Provide a NUMBERED LIST of the TOP 3 5  most critical issues.
- Each item MUST follow this exact format:
  [CATEGORY] Issue Description -> Actionable Fix
- Categories must be chosen from:
  [BOUNDARIES], [CONNECTORS], [TEXT], [ALIGNMENT], [STYLE]
- Negative Constraints:
  - DO NOT output Python code.
  - DO NOT give vague or generic advice.

Example Output:
1. [BOUNDARIES] The 'Output' block on the far right is clipped -> Shift the 'Output'
    block and its label LEFT by approximately 1 inch.
2. [CONNECTORS] The arrow from 'Encoder' to 'Decoder' crosses the text -> Change the
    connector type to Elbow.
3. [CONNECTORS] Arrowheads on the main pipeline are too large and obscure text ->
   Reduce arrowhead size to Medium.
4. [TEXT] The 'Feed Forward' label is light gray -> Change font color to BLACK.
"""
```

