# OpenReview forum: "LiveFigure: Generating Editable Scientific Illustration with VLM Agents"
_ICML.cc/2026/Conference — ICML 2026 regular_

### Official Review · Reviewer_c2sH · 2026-03-12

**Soundness:** 3
**Presentation:** 2
**Significance:** 4
**Originality:** 4
**Overall Recommendation:** 4
**Confidence:** 5

**Summary:**

This paper presents a method to address an editability limitation of existing LLM-based approaches for generating figures for scientific papers. Because such methods do not produce editable outputs, even minor revisions often require regenerating the entire figure, and fine-grained modifications are difficult to make.

Instead of following the conventional prompt-to-pixel paradigm, the paper proposes agents that performs figure generation in multiple stages, in a way that more closely resembles how humans create figures in practice. The evaluation is conducted on sampled figures from prior papers, and performance is reported using nine metrics as well as edit distance.

**Compliance With Llm Reviewing Policy:**

Affirmed.

**Final Justification:**

Although the paper still appears to have room for further development in light of the other reviews, the authors’ follow-up addressed most of my concerns, so I maintain my current positive score.

**Key Questions For Authors:**

1) How is publication readiness determined?
2) How are skills selected from the Skill Library depending on the situation, and what policy is used for that selection?
3) How were tools such as Graphviz and Matplotlib evaluated?
4) Since the paper appears to use the NanoBanana as its backbone, is it appropriate to compare the proposed method against NanoBanana?

**Limitations:**

This paper doesn't adequately consider the limitations of the methodology itself. It seems necessary to discuss the potential for failures in the test results and the risk of producing a picture that appears sophisticated but is scientifically inaccurate.

**Strengths And Weaknesses:**

The main strength of this work is that it successfully addresses the lack of editability in prior methods for scientific figure generation by reformulating the problem as a procedural, agentic system. It also defines edit distance and uses it to evaluate the results in an appropriate way.
That said, the paper does not sufficiently explain how publication readiness is determined, or how tools such as Graphviz and Matplotlib are evaluated. Clarifying these aspects would likely improve confidence in the reported results.
In addition, the paper does not provide evaluations for each stage of the three-stage agent pipeline. Since the paper claims that the agent is designed to mimic the way humans create figures, it may be worthwhile to evaluate the output of each stage separately in order to better support that claim. If quantitative evaluation is difficult, even a qualitative analysis could be helpful.
One additional point I am curious about is how the system selects appropriate skills from the Skill Library depending on the situation. It appears that the model implicitly invokes the relevant skills, but it would be helpful if the paper explained how the model is guided to choose the right skills in the right context.

The paper is also concise and generally well written in terms of its overall structure, particularly in how it presents the problem setting, method, and results. However, the terminology and clarity could be improved. For example, multiple names such as “Gemini-3-Image-Pro,” “Gemini-3-Pro-Image,” and “Gemini Nano Banana” are used somewhat interchangeably, which can be confusing. In addition, the typo “NanoBanna” appears repeatedly throughout the paper.

---

> ### Author Rebuttal · Authors · 2026-03-31
>
> Thank you very much for your insightful reviews of our work. We address your questions as follows:
>
> **Criteria for "Publication Readiness" (W1 & Q1)**
>
> "Publication Readiness" is strictly operationalized through two quantifiable dimensions:
> - **Human Evaluation (Edit Distance):** We measured the manual effort required to polish model outputs until **7 AI researchers with PhD degree** (each with 3+ top-tier publications) confirmed the figures reached "publication-ready" standards. As both frequent creators and reviewers of high-level papers, their consensus authentically reflects professional academic standards. "Edit Distance" thus objectively quantifies the specific intervention steps needed to transform an initial draft into a final "publication-ready" state.
> - **VLM-as-a-judge Evaluation:** We mapped official guidelines from **Nature, IEEE, and NeurIPS** into a structured rubric for a GPT-4o "Senior Reviewer" across three dimensions:
>     - **Dimension 1: Visual Design.** Based on _Nature_ [1, 2] and _IEEE_ [5] standards; we strictly penalize text overlap, superfluous elements, and inconsistent spacing.
>     - **Dimension 2: Information Clarity.** Following _NeurIPS_ instructions [4] requiring artwork to be "neat, clean, and legible," we evaluate the unambiguity of logical flow and text clarity.
>     - **Dimension 3: Content Fidelity.** Anchored in _Springer Nature_ publishing ethics [3], our "Accuracy" metric dictates severe penalties for hallucinated relationships or incorrect topological logic.
> By translating these real-world norms into 9 quantitative metrics, we achieve a standardized and objective scoring of "publication readiness." Full criteria and citation URLs [1-5] are provided here:
> https://anonymous.4open.science/r/LiveFigure/Rebuttal/Publishability_Criteria.png
>
>
> **Evaluation Details of Tools (W2 & Q3)**
>
> - First, the inputs for these code-based baselines were entirely identical to those used for LiveFigure, utilizing either Caption-only (V1) or Caption + Method Description (V2).
> - Second, these baselines were generated directly by Gemini 3 Pro. We prompted the model to output the complete Graphviz (DOT language) or Matplotlib (Python script) code based directly on the input text descriptions. Subsequently, the generated code is automatically compiled and executed in a local sandbox environment, enabling direct rendering and export of PNG or PDF images.
> - Finally, the evaluation method was identical to that of LiveFigure. The generated diagrams were submitted to the VLM judge and evaluated across the exact same 9 dimensions in a double-blind manner.
>
>
> **Skill Selection Strategy (W4 & Q2)**
>
> LiveFigure leverages the **Multimodal In-context Learning** and code reasoning capabilities of VLMs to achieve flexible skill invocation through a structured three-step process:
> - **API Documentation Injection:** The skill library $\mathbb{S}$ is provided to the Coding Agent as a concise manual within the system prompt, containing functional descriptions, parameter constraints, and usage examples for each standardized Python API.
> - **Autonomous Multimodal Mapping:** The Agent receives both the user’s text and the **visual blueprint image**. It utilizes native multimodal understanding to map visual elements in the sketch directly to specific APIs. For instance, it invokes `add_container` for identified logical groupings, `add_connector` for directional topological links, and `add_text_node` for textual content.
> - **Spatial Planning & Execution:** The Agent estimates coordinates $(x, y)$ and dimensions for each module, then plans a chronological execution order (e.g., instantiating background containers before child nodes and connections). This ensures correct layering and obstacle-avoidance logic during rendering.
> This mechanism effectively simulates a human programmer’s workflow: "Analyzing requirements $\rightarrow$ viewing layout sketches $\rightarrow$ consulting API docs $\rightarrow$ writing structured code."
>
> **Nano Banana as a Baseline (Q4)**
> Comparing with Nano Banana is appropriate and necessary, as it represents the current SOTA for end-to-end image generation and the "prompt-to-pixel" paradigm. This comparison demonstrates a core premise: even with exceptionally strong image models, single-pass raster images cannot meet the rigorous demands of scientific publishing for vectorization, editability, and structural clarity. As shown in our Edit Distance experiment (Fig. 3), Nano Banana achieves the highest initial acceptance rate (24%), but this acts as a hard ceiling. In contrast, LiveFigure matches this rate within just 2 editing steps and significantly surpasses it thereafter. Thus, the comparison proves that our approach is the superior path for addressing scientific illustration.
>
> **Due to space constraints, we could not fit all our responses in this rebuttal. Please feel free to reply if you would like us to provide the remaining details in a follow-up.**

---

> > ### Author Rebuttal · Reviewer_c2sH · 2026-04-03
> >
> > Thank you for the detailed rebuttal. The clarifications on how publication readiness is operationalized, how the Graphviz and Matplotlib baselines were run and evaluated, and how skills are selected from the library are all helpful. I also agree that Nano Banana is a relevant baseline for comparing an editable pipeline against a strong prompt-to-pixel model.
> >
> > However, one of my original concerns remains: the paper still does not provide stage-wise evaluation or qualitative diagnosis of the three-stage pipeline. Since the central claim is that the system mirrors how humans iteratively construct scientific figures, I still think the paper would be much stronger with either quantitative results or focused qualitative analysis for each stage. Without that, it is harder to tell where the gains are actually coming from and which parts of the pipeline are most responsible for editability and final quality.
> >
> > I also still encourage a more explicit discussion of failure modes, especially cases where the figure appears polished but is scientifically inaccurate. The publication readiness rubric is useful, but it does not fully substitute for an analysis of scientific correctness failures. Finally, the terminology inconsistencies around the backbone naming and the repeated NanoBanna typo should be fixed in the revision.
> >
> > Overall, the rebuttal addresses several important questions, but my concerns are only partially resolved, and I would maintain my current evaluation.

---

> > > ### Author Response · Authors · 2026-04-04
> > >
> > > Thank you for the follow-up. Due to the character limit in the initial rebuttal, we omitted the detailed stage-wise evaluation and specific failure analyses, which we now provide.
> > >
> > > #  Quantitative Stage-wise Evaluation
> > >
> > > We have conducted a comprehensive stage-wise evaluation. We sequentially added each stage of our pipeline to a vanilla LLM baseline, tracking the Executability, Debug Turns, and final VLM Score. The quantitative progression is presented in the table below:
> > >
> > > | **Stage**                           | **Executability ↑** | **Debug Turns ↓** | **VLM Score ↑** |
> > > | ---------------------------------------------------- | ----------------- | ----------------- | --------------- |
> > > | **Vanilla LLM Generation**                           | 31.0%             | 3.4               | 5.03            |
> > > | **Row 1: + Stage 1 (Visual Prior)**                  | 32.5%             | 3.5               | 5.75            |
> > > | **Row 2: + Stage 2 (Skills & Experience)**           | 83.5%             | 0.46              | 6.78            |
> > > | **Row 3: + Stage 3 (Refinement) -> Full LiveFigure** | 83.5%             | 0.53              | 7.28            |
> > >
> > > **Stage-wise Diagnosis & Analysis:** This quantitative breakdown explicitly demonstrates how each module incrementally contributes to the system's **efficiency** (measured by Executability and Debug Turns) and **generation quality** (measured by VLM Score):
> > >
> > > - **Row 1 to Row 2 (Visual Planning Gain):** Introducing the Stage 1 Visual Prior provides the system with human-like macro-layout planning. Consequently, the VLM Score improves from 5.03 to 5.75 due to more logical spatial arrangements. However, Executability remains low (32.5%), and Debug Turns slightly increase (from 3.4 to 3.5). This authentically reflects that forcing a vanilla LLM to faithfully render a complex visual blueprint using raw, unconstrained Python code actually increases the coding difficulty, leading to more trial-and-error.
> > > - **Row 2 to Row 3 (Execution Efficiency Improvement):** Introducing Stage 2 (Skill Library & Debug Experience) completely resolves the aforementioned execution bottleneck. Executability skyrockets from 32.5% to 83.5%, while Debug Turns plummet to 0.46. By encapsulating rendering logic into structured APIs, the system translates abstract layouts into functional code almost in a single pass. This massively improves execution efficiency and boosts the VLM Score to 6.78, as the planned layouts are now accurately and reliably rendered.
> > > - **Row 3 to Row 4 (Publication Quality Polish):** Stage 3 acts as the visual micro-aligner. In this stage, the Agent executes targeted, minor parameter adjustments based on visual feedback to address subtle imperfections in spatial layout, graphic rendering, and element attributes exposed in the initial draft. This highly focused refinement effort significantly elevates the VLM Score to our final SOTA performance of 7.28, quantifying its indispensable role in bridging the gap to publication-ready quality.
> > >
> > > # Analysis of Failure Modes
> > >
> > > We examined the generated illustrations in the test set and provide a specific failure case here:
> > > > https://anonymous.4open.science/r/LiveFigure/Rebuttal/failure-logic.png
> > >
> > > This generated flowchart for the paper _"Foundation Model-oriented Robustness: Robust Image Model Evaluation with Pretrained Models"_ appears visually neat and structured but contains scientific inaccuracies:
> > > - **Logical Omission (Component Isolation):** The "Surrogate Oracle Ensemble h" block is visually well-placed but lacks a logical connective arrow to the "Oracle Validation: $h(\hat{x}) = y?$" decision node, breaking the algorithmic data flow.
> > > - **Core Concept Deviation:** In the bottom examples (Fig B), the Agent explicitly morphed a "Dog" into a "Wolf", and a "Stop" sign into a "Yield" sign.
> > >
> > > These errors expose the current limitations in agent reasoning when processing dense scientific logic. For the topological omission, the model prioritized overall visual harmony while neglecting detailed data dependency logic. More fundamentally, the conceptual deviation in Fig B indicates that the LLM failed to deeply comprehend the mathematical premise ($h(\hat{x}) = y$) during the reasoning phase. Instead of understanding that a "perturbation" must preserve the core semantic identity (e.g., adding noise or texture variations to a dog), the model superficially interpreted the task. It resorted to a simplistic, text-level keyword replacement: simply swapping discrete entity names (from "Dog" to "Wolf") to represent a "perturbed" state.
> > >
> > > We will include a dedicated section with more detailed analyses and discussions of specific failure cases in the revised manuscript.

---

### Official Review · Reviewer_FKAn · 2026-03-13

**Soundness:** 2
**Presentation:** 2
**Significance:** 3
**Originality:** 2
**Overall Recommendation:** 4
**Confidence:** 3

**Summary:**

The authors propose LiveFigure, a framework for generating figures for scientific illustration with VLM agents. Designed in a precoedural way, the proposed approach involves three phases. Initially, given the instruction for generating the requested illustration, the agents conduct a planning stage, and extend the asset library as requested. In this planning stage, the authors utilize the past papers from top venues (e.g. NeurIPS, ICML, ICLR) as visual priors. Following this phase and the generated plan, the framework undergoes the procedural generation phase, which utilizes a predefined skillset. As the final steps, the proposed framework enables the capability of refining the powerpoint for structural errors. Overall, the proposed method enables a generation mechanism for scientific illustrations suitable for scientific papers, that are actionable (in PowerPoint format) and differing from static images. The authors demontstrate the effectiveness of their method with an human oriented evaluation protocol.

**Compliance With Llm Reviewing Policy:**

Affirmed.

**Final Justification:**

The authors have succesfully addressed my concerns regarding the terminology, cost, and details on the distribution and evaluation of the examples. With respect to the authors rebuttal, I am more leaning towards acceptance of the paper, which is an useful framework for utilzing multi-agent frameworks for scientific figure generation. Now that the experimental protocol and the terminology is more clear, I have revised my score from weak reject to weak accept.

**Key Questions For Authors:**

- How many samples does the method require to construct $\mathbb{K}$? The authors provide the details for collecting the set, but without the information regarding the set size. Also for this specific domain (scientific figures), does the performance change based on the prior examples? The answers to this questions would be informative to understand the manual work required for the method.
- How extendible is the method to vision-oriented papers? For the context of visual-generation oriented papers, the figures may rely on the use of visual examples from the papers. Such examples constitute to figures that are not qualitatively represented in the paper. Can the proposed method generalize to such generations? Does the filtering of non-schematic figures prevent such generations?
- Can the authors provide more details on the skill library $\mathbb{S}$, while it makes sense that pre-defined operations in powerpoint helps modelling the figure, the operations that are supported is an important detail in the framework and would be a good indicator that how easily adaptable LiveFIgure to other file formats. Are these operations domain specific or diagram-type specific?
- For the evaluations, what is the number of participants included? Is the number of individuals involved in the assessment sufficient for statistical significance?

Please also see the weaknesses section. I would be willing to increase my score in case these concerns are addressed.

**Limitations:**

The authors do not provide an explicit limitations section, to understand the extent of the paper both the usage cost (constructing the priors and the runtime cost), and the generation limitations is suggested to be mentioned. Please see the questions and weaknesses sections for further details. For the broader impact statement, the included acknowledgement is sufficient.

**Strengths And Weaknesses:**

**Strengths**
- The proposed method serves as a good use case of VLM agents for a challenging use case, where the hierarchy of the diagram elements are not straight-forward.
- The experimental section is strong. Authors include comparisons with state-of-the-art multimodal backbones, and report the gains they obtain both in comparison with the used backbone and the other models.
- Visual examples of the generated illustrtations are convinving for the quality of the generations.
- The paper extends the discussion for scientific illustration generation from only generation to generation and usability.

**Weaknesses**
- In the terminology of the paper, some of the variables and sets are defined in a vague way. As an example $\mathcal{B}$ is defined as a visual blueprint of the illustration to be generated, but the details and the format of it is not precisely defined. For $\mathbb{A}$ it is more understandable but ideally all of these sets should be explained further for clarity.
- In the evaliation baselines, the number of human experts that are editing, judging the results along with the participants of the Human Eval is not defined. Consequently we only have information that they are experts, but details on their domains of expertise along with the domains of the papers included in the studies are now provided. For example, domains of similar nature are expected to have similar diagrams and the samples won't be reprsentative in terms generalization to all scientific illustrations.
- The method provides self correction and itartive refinement loops first to generate a actionable file and refining the visual details. What is the impact of this in the token cost of the agentic pipeline. A comparison on this would be important on the cost of the overall procedure. In addition, ablations on the correction and refinement steps (number of steps) is missing.
- Typo in L44, L46 of main paper: NanoBanna -> NanoBanana
- The comparisons with the methods such as TikZ, HTML/CSS (highlighted with light blue) raises concerns. The procedure of generating the figures is not explained in detail. If they are generated with LLMs directly, this would not be fair as they cannot utilize the multi-agent frameworks and the gains from  an agentic orchestration would be expected. If these comparisons utilize the same flow but only replaces the PowerPoint decision, that should be acknowledged too.

---

> ### Author Rebuttal · Authors · 2026-03-31
>
> Thank you for your constructive suggestions. We address your concerns point by point as follows:
>
> **Definition of Variables (W1)**
> We provide further specifications for the variables and sets mentioned:
> - **Visual Blueprint $\mathcal{B}$:** Represented as an **image file**. As shown in Fig. 2, $\mathcal{B}$ is a spatial map generated by the Blueprint Agent that defines the macro-layout, coordinate constraints, and topological relationships using color blocks and wireframes. It serves as the primary visual reference for subsequent code generation.
> - **Asset Library $\mathbb{A}$:** Defined as a **collection of independent images** (e.g., icons, rendered examples) generated via the image model. Each asset in $\mathbb{A}$ is a cropped, background-removed image file that the Coding Agent precisely anchors to specific coordinates within the final vector graphic.
>
>
> **Expert Background & Domain Distribution (W2 & Q4)**
> - **Human Evaluation:** We invited **7 AI researchers with PhD degree**, each with 3+ top-tier publications, to evaluate the outputs. As both target users and expert reviewers for venues like ICML/NeurIPS, their expertise perfectly aligns with our test set. In the double-blind preference tests (Fig. 5), LiveFigure’s win rates strictly passed the **95% Confidence Interval test ($p < 0.05$)**, confirming the results are statistically significant and reliable.
> - **Domain Distribution:** We counted the domain distribution of the 300 papers in the test set as follows:
>   https://anonymous.4open.science/r/LiveFigure/Rebuttal/domain_distribution.png
>
>
> **Cost Analysis and Refinement Ablation (W3)**
> Based on test set statistics, we break down the costs and the impact of the iterative process:
> - **Total Cost:** Average end-to-end cost per figure is **~$0.80** (~90s).
>     - **Image Gen:** ~2 calls ($0.38).
>     - **VLM:** ~12 calls ($0.42).
> - **Incremental Token Burden:**
>     - **Base Cost (Single-pass):** Includes 2 image calls and 6 base VLM calls, totaling **~$0.59**.
>     - **Self-Correction:** Each debug round (code repair via error logs) adds **1 VLM call (~$0.035)**.
>     - **Visual Refinement:** Each visual feedback round (layout fine-tuning) adds **2 VLM calls (~$0.07)**.
> - **Ablation Results:** Fig. 4 in the manuscript demonstrates steady gains in **Design, Clarity, and Fidelity** across iteration steps. Removing the refinement module entirely (`w/o Refinement` in Table 2) causes the VLM Score to drop from **7.28 to 6.78**.
> Compared to hours of manual drafting, this minimal cost increment is justified by the significant gains in executability and visual quality.
>
>
> **Implementation Details of Code Baselines (W5)**
>
> We clarify the implementation and rationale for our code-based baselines:
> - **Standard Baselines (Direct Generation):** Baselines like TikZ and HTML/CSS were generated directly by **Gemini 3 Pro**. The purpose of this comparison is to demonstrate the systemic performance gap between the "direct code generation" paradigm and our proposed **agentic orchestration paradigm**. Indiscriminately applying our multi-agent framework to all baselines would obscure the independent value of the framework itself.
> - **Ablation Study (Code Representation):** To address your suggestion, we conducted an experiment by applying the complete LiveFigure workflow to other formats (50 random tasks, V1 input). This transforms the comparison into an ablation of the **"Code Representation"** (PPTX vs. Others). The results are as follows: https://anonymous.4open.science/r/LiveFigure/Rebuttal/LiveFigure+Code.png
> *Note: "Improvement" is relative to direct generation without the LiveFigure framework.
>
> While the LiveFigure workflow improves all formats, they still significantly lag behind our **PPTX-based representation**, validating our choice of underlying data structure.
>
>
>
> **Extensibility to Vision-Oriented Illustrations (Q2)**
>
> LiveFigure is highly adaptable for vision-oriented illustrations requiring specific experimental results or biological assets. Users can either provide ready-made assets (e.g., screenshots) or describe them during the generation phase. The Agent then incorporates these into the **Visual Planning** layout and anchors them precisely during the coding phase.
> To demonstrate this extensibility, we added two cases:
> - **Computer Vision (3D Asset Acquisition):** https://anonymous.4open.science/r/LiveFigure/Rebuttal/vision-oriented.png
>   This diagram illustrates a multi-view RGB-D capture and mesh post-processing workflow. LiveFigure successfully performed spatial layout and logical routing for user-described assets like human models and texture maps.
> - **Biology (Exosome Purification):** https://anonymous.4open.science/r/LiveFigure/Rebuttal/biology-example.png
>   This case demonstrates the separation of Extracellular Vesicles (EV), incorporating several specialized biological visual elements.
> These prove that LiveFigure generalizes effectively to **vision-dominated / cross-disciplinary** illustrations.

---

> > ### Author Rebuttal · Reviewer_FKAn · 2026-04-04
> >
> > Thanks to the authors for the responses. Most of my concerns has been resolved with the clarifications provided. I have adjusted my score accordingly. As a additional question, that would be appreciated if can be answered, how many samples does it take to construct the initial set $\mathbb{K}$? It would be appreciated if sample set sizes can be shared.

---

> > > ### Author Response · Authors · 2026-04-04
> > >
> > > Thank you for your timely feedback and for the score adjustment! Regarding the construction of the initial set $\mathbb{K}$, we are happy to share the details:
> > > - **Initial Sample Size:** Our initial Knowledge Base $\mathbb{K}$ consists of **600 "illustration-text" pairs** automatically curated from papers accepted at ICLR 2025, NeurIPS 2025, and ICML 2025.
> > > - **Fully Automated Scaling & Generalization:** The construction of $\mathbb{K}$ is fully automated through a two-stage pipeline. It first uses rule-based parsing to extract illustration figures and captions from PDFs, followed by VLM-driven semantic filtering to retain only high-quality methodological illustrations while discarding noise (e.g., experimental plots). This automated approach requires zero manual annotation, allowing LiveFigure to continuously scale by incorporating the latest publications. It also enables seamless generalization to other domains, such as schematic diagrams in biology.
> > > - **Commitment to Open Access:** We are fully committed to the spirit of open science. Upon acceptance, we will open-source the complete dataset along with our automated construction pipeline to help the community easily customize and extend "AI Scientists" for various research fields.
> > >
> > > We hope these clarifications address your question, and we welcome further discussion.

---

### Official Review · Reviewer_NswF · 2026-03-19

**Soundness:** 3
**Presentation:** 3
**Significance:** 2
**Originality:** 2
**Overall Recommendation:** 4
**Confidence:** 3

**Summary:**

To address the challenge of existing image generation models in producing high-quality, editable scientific illustrations, this paper proposes an AI Agent framework called LiveFigure. Based on an advanced VLMs, it simulates the multi-step drawing process of human experts, generating executable code through three stages: Visual Planning, Procedural Generation, and Targeted Refinement. The final editable scientific illustration is then generated using PowerPoint. Furthermore, the reliability of the code is ensured through a predefined standard skill library and self-correction techniques. Experiments show that this framework can generate higher-quality scientific illustrations than general image generation models and traditional code libraries with a limited number of editing steps.

**Compliance With Llm Reviewing Policy:**

Affirmed.

**Final Justification:**

4: Weak accept

**Key Questions For Authors:**

1. While many quantitative metrics and human assessments are provided, how can the relevance of these metrics to the publishability criteria mentioned in the paper be verified?

2. The framework heavily relies on closed-source LLM. How much of this performance improvement comes from the pipeline design, and how much from the capabilities of Gemini/GPT itself?

3. What is the cost of generating each image, how many API calls are required, how many tokens are consumed, and how much time is spent?

4. Can you provide some examples from other disciplines, such as biology?

**Limitations:**

The authors did not discuss the limitations of this work, but they could provide some case studies of failures. The system first retrieves relevant documents and then generates a visual blueprint and some entity elements. Is there a risk of copying content from the retrieved documents, such as icons, at this step?

**Strengths And Weaknesses:**

Strengths:

1. Generating editable scientific illustrations using code and existing toolkits (PowerPoint) overcomes the limitation of general generative models that can only generate raster images, making it more suitable for practical use.

2. The constructed LiveFigure framework is complete, and the three-stage pipeline design appears reasonable. Techniques such as prior knowledge retrieval, a standard skill set, and iterative refinement ensure the system's reliability.

3. The paper is clearly written, with complete figures and tables. Qualitative and quantitative experiments validate the framework's effectiveness.

Weaknesses:

1. While LiveFigure can directly generate editable PPT files, it seems to lack interactive modification support; users cannot instruct the system to modify existing files.

2. Although the paper uses many evaluation metrics, there is no detailed explanation of these metrics, and it's unclear how they are calculated. If VLM is used for evaluation, the model used, the system prompt used, and any biases between different VLMs should be explained.

3. As a usable agent system, it lacks cost analysis, including LLM token consumption and the time required to generate a single scientific illustration.

4. The results in Table 1 show that LiveFigure does not have a significant advantage over Nano Banana and gpt-image-1.5, and even in human evaluation, the advantage over gpt-image-1.5 is not significant (60 and 40). More detailed information about the human evaluation should be provided, such as the number of evaluators and their knowledge background (since the evaluation of scientific illustrations is involved).

---

> ### Author Rebuttal · Authors · 2026-03-31
>
> Thank you for your constructive feedback. Below is our detailed response addressing your concerns.
>
>  **Interactive Modification (W1)**
>
> We clarify that LiveFigure **natively supports language-based interactive modification** due to our code-generation paradigm. Unlike raster models that require full-image regeneration for minor tweaks, our Coding Agent can precisely locate and modify specific code segments corresponding to independent objects. This ensures that modifications (e.g., "Change the Encoder to blue") strictly preserve the original layout and topology.
> To demonstrate this, we added a sequential case study:
> 1. Original Figure: https://anonymous.4open.science/r/LiveFigure/Rebuttal/before_edit.png
> 2. **Instruction 1:** _"Recolor rounded rectangles in the 'closed-loop' block to light green."_ Output: https://anonymous.4open.science/r/LiveFigure/Rebuttal/after_edit_1.png
> 3. **Instruction 2:** _"Delete the bottom title and shift the 'tree search' block inward to fit the background."_ Output: https://anonymous.4open.science/r/LiveFigure/Rebuttal/after_edit_2.png
> Both natural language edits were executed successfully.
>
>
> **Detailed Calculation of Evaluation Metrics (W2)**
>
> We provide the following clarifications regarding our evaluation protocols:
> - **Edit Distance (Fig. 3):** Calculated via an automated PPTX parsing script that directly **diffs the underlying XML trees** before and after manual editing. It objectively tallies atomic operations (Add, Delete, Modify) on nodes and attributes to yield the final distance.
> - **VLM-as-a-judge (Table 1):** We utilized GPT-4o as the judge. The System Prompt assigned the role of "Senior Scientific Reviewer" and mandated a structured rubric across 9 metrics. Full prompt details are available here: https://anonymous.4open.science/r/LiveFigure/Rebuttal/vlm_prompt.png
> - **Ablation Study (Table 2):** Executability measures the success rate of bug-free code compilation. Debug Turns records the iterative rounds performed by the Coding Agent. VLM Score is the mean of the 9 VLM-as-a-judge metrics.
>
>
> **Cost and Efficiency Analysis (W3 & Q3)**
>
> Based on test set statistics, the average cost and time per scientific figure are:
> - **VLM Usage:** \~12 calls (150K Input / 10K Output tokens). At standard pricing ($2/1M Input; $12/1M Output), cost is **~$0.42**.
> - **Image Generation:** \~2 calls (2K Input / 3.15K Output tokens). At standard pricing ($2/1M Input; $120/1M Output), cost is **~$0.38**.
> - **Total Efficiency:** The end-to-end workflow takes **~90 seconds**, with a total average cost of **~$0.80 per figure**.
> Compared to the hours of manual labor typically required for publication-quality drafting, LiveFigure offers a highly cost-effective and scalable solution for researchers.
>
>
>
> **Publishability Criteria (Q1)**
>
> "Publishability" is defined via:
> - **Human Edit Distance:** XML-level effort (Add/Del/Mod) for 7 AI researchers with PhD degree to reach "ready" standards.
> - **VLM-as-a-judge:** 9 metrics mapped from **Nature/IEEE/NeurIPS** [1-5] via GPT-4o covering Design, Clarity, and Fidelity. Citation URLs [1-5] are detailed here:
> https://anonymous.4open.science/r/LiveFigure/Rebuttal/Publishability_Criteria.png
>
>
> **Attribution of Performance Improvement (Q2)**
>
> While Gemini provides the foundational reasoning, the substantial performance gain comes from our **pipeline**. Generating editable scientific figures is a long-horizon task spanning text comprehension, spatial planning, code execution, and visual feedback—tasks vanilla LLMs cannot handle in one pass.
> **New Global Ablation Study:** To further quantify the pipeline’s contribution, we removed all designed mechanisms (Experience + Skills + Refinement + Visual Prior) and prompted the vanilla model directly. Executability dropped to **31.0%**, and visual quality fell to **5.03**, a **30.9% decrease** from the full pipeline. Previous ablations also show that removing individual components, e.g., "Experience-Driven Constraints," reduces executability to 40.0% (Table 2). This demonstrates that while LLMs provide reasoning capability, LiveFigure’s system architecture is essential for reliably generating editable scientific figures.
>
>
> **Cross-disciplinary Example (Q4)**
>
> We have generated an example from the biology domain:
> https://anonymous.4open.science/r/LiveFigure/Rebuttal/biology-example.png
> https://anonymous.4open.science/r/LiveFigure/Rebuttal/biology-example-pptx-ui.png
> This diagram illustrates the separation and purification workflow of Extracellular Vesicles.
>
> **Failure Cases:**
>
> Although LiveFigure performs robustly in the vast majority of cases, it occasionally struggles with overly crowded modules or "spaghetti routing" when processing complex system architectures with extremely high information density. We have provided a failure case analysis here: [https://anonymous.4open.science/r/LiveFigure/Rebuttal/Failure_Case.png](https://anonymous.4open.science/r/LiveFigure/Rebuttal/Failure_Case.png)

---

> > ### Author Rebuttal · Reviewer_NswF · 2026-04-03
> >
> > My concerns have been adequately addressed.

---

> > > ### Author Response · Authors · 2026-04-04
> > >
> > > We sincerely thank you for your positive feedback. We are glad that our additional experiments and clarifications have addressed your initial concerns.
> > >
> > > Given your assessment that the concerns are adequately resolved, we would greatly appreciate it if you would consider reflecting this in your final score, as it may help the Area Chairs in their decision-making process.
> > >
> > > Thank you again for your time and constructive insights.

---

### Official Review · Reviewer_SLPQ · 2026-03-24

**Soundness:** 3
**Presentation:** 3
**Significance:** 3
**Originality:** 3
**Overall Recommendation:** 3
**Confidence:** 4

**Summary:**

This paper proposes LiveFigure, an agentic framework for generating editable scientific figures (PPTX vector graphics). The method decomposes figure generation into three stages: visual planning via prior induction, procedural generation via code and skill libraries, and iterative refinement via visual diagnostics. The key idea is to shift from pixel-based generation to code-driven procedural construction, and to evaluate editability using Edit Distance. Experiments show improved editing efficiency and human preference over existing baselines. The work is practically valuable and well-engineered, but methodologically incremental. Stronger novelty and more comprehensive comparisons would improve its impact.

**Compliance With Llm Reviewing Policy:**

Affirmed.

**Final Justification:**

I appreciate the authors’ effort in addressing the reviewers’ comments. However, I still find that the technical contribution of the work, particularly in terms of the agentic orchestration and system design, is not yet sufficiently well-justified or clearly distinguished from existing practices. While the proposed system is well-engineered, it remains unclear to what extent it introduces fundamentally new technical insights beyond existing approaches for building similar pipelines. In particular, the paper would benefit from a more explicit characterization of what novel principles or design choices lead to improved capability, rather than relying primarily on system integration. Overall, I believe that these concerns would require a more substantial revision to address, and I will maintain my original score for the current submission.

**Key Questions For Authors:**

1. With the rapid progress of image generation models (e.g., Gemini) that can produce high-quality scientific figures with strong content understanding and text accuracy, is it better to convert raster outputs into vector graphics, or to generate figures natively via code-based procedural methods? There seems to be no systematic comparison between these two approaches—what are their respective strengths and weaknesses?

2. For metrics that are hard to quantify—such as correctness and interpretability of figures—can we use a VQA-style evaluation? Specifically, an external VLM is asked to interpret the generated figure and describe its content. By comparing this understanding with the intended meaning, we can assess whether the figure accurately conveys the user’s intent.

**Limitations:**

However, my main concern is on the technical side: the method appears to be largely an engineering integration, and the level of technical depth may not be sufficient for ICML. This could be acknowledged as a limitation. Additionally, from the perspective of result quality, the visual outputs are not particularly impressive compared to some recent works.

**Strengths And Weaknesses:**

Strengths

1. Practical and well-motivated problem
The paper addresses the important limitation that generated figures are not editable, and introduces a concrete metric (Edit Distance) to quantify usability.

2. Insightful paradigm shift
Moving from raster generation to procedural generation, with a multi-stage cognitive pipeline, is conceptually clear and meaningful.

3. Strong system design
The combination of skill libraries, experience injection, self-correction, and visual feedback forms a robust and effective pipeline.

Weaknesses

1. Limited methodological novelty
The approach mainly integrates existing components (VLMs, retrieval, code generation, refinement loops) with limited new algorithmic contribution.

2. Evaluation limitations
 The evaluation metrics exhibit a certain degree of subjectivity, and the reliability of the Visual Evaluation Metrics remains unclear. Edit Distance also has inherent limitations, and the human evaluation is relatively small-scale with insufficient statistical significance. Therefore, the claims regarding “practical usability” require stronger support.

---

> ### Author Rebuttal · Authors · 2026-03-31
>
> We sincerely thank you for the constructive feedback. We address each of your concerns below.
>
> **Novelty (W 1 & Limitations)**
> We clarify our core contributions from three perspectives:
> 1. **Composite Innovation Addressing a High-Value Problem:** First, we formally identify and mathematically define a previously overlooked yet pervasive problem—generating _natively editable_ scientific vector figures. To address this, we propose a novel agentic orchestration solution bridging semantic text and structured visual code.
> 2. **Agentic Workflow Orchestration as a Paramount Innovation Domain:** Second, with the rapid commoditization of foundational LMs, the research and design of Agentic Workflows have emerged as a critical domain of technical innovation. Transforming a raw LM into a reliable, complex workflow is a profoundly non-trivial engineering and algorithmic challenge. In our early explorations, we found that directly using LMs frequently suffers from severe API hallucinations and suboptimal spatial reasoning. Therefore, LiveFigure is far from a simple concatenation of APIs. The recent success of OpenClaw further demonstrates that orchestrating models into functional, robust skills is driving the next wave of AI capabilities.
> 3. **Alignment with the ICML "Applications" Scope:** Finally, the primary area of our submission is "Applications." According to the official ICML Call for Papers, the scope explicitly welcomes _"application-driven machine learning (innovative techniques, problems, and datasets that are of interest to the machine learning community and driven by the needs of end-users)."_ LiveFigure perfectly aligns with this objective.
>
>
> **Reliability of Evaluation Metrics (W 2 & Q 2)**
> Our evaluation combines objective automated metrics and structured human evaluation, supported by strict statistical significance testing.
> - **Objective Alignment with SOTA Protocols:** First, our VLM-as-a-judge metrics follow protocols from recent works like **AutoFigure (ICLR 2026)**, ensuring automated scoring is consistent with current academic standards.
> - **Structured Human Evaluation:** Sole reliance on VLM metrics risks "metric hacking" and misses our contribution: the _editability_ and _practical utility_ of figures. Human evaluation is therefore indispensable. To minimize the variance typically caused by human subjectivity, we implemented a highly systematized protocol. For instance, "Edit Distance" is computed by diffing XML source codes of PPTX files before and after human edits, filtering out human bias and measuring actual effort to reach a publication-ready state.
> - **Statistical Significance:** LiveFigure’s win rates against all baselines in double-blind human tests passed 95% confidence interval tests ($p < 0.05$), demonstrating stable and reliable advantages.
>
> Following your suggestion, we added an experiment on image information retention. Generated illustrations were interpreted by a VLM (GPT-4o), and GPT-5 compared these descriptions with original captions and methods, assigning an "Intent Conveyance Score" (0–5) based on method completeness and logical correctness. Results (https://anonymous.4open.science/r/LiveFigure/Rebuttal/retention.png) confirm LiveFigure’s advantages in figure "Information Clarity" and will be included in the revised manuscript.
>
>
> **Comparison with other generation approaches (Q 1)**
> Thank you for this highly constructive question. We provide a detailed comparative discussion as follows:
> - **Raster-to-Vector Pipeline:** This approach involves generating a raster image first and then converting it into vector format via edge detection and color patch clustering.
>     - **Pros:** High retention of complex visual effects (e.g., lighting, gradients, textures).
>     - **Cons:** **Fundamental lack of semantic editability.** Vectorization shatters text into meaningless curve paths rather than editable strings and clusters logical arrows into irregular polygons. Overlapping layers are often forcibly flattened, making any subsequent adjustment (e.g., fixing a typo or moving a module) a disastrous manual experience.
> - **Native Code-based Procedural Generation:** This pathway constructs graphics programmatically via executable scripts.
>     - **Pros:** **Genuine semantic editability and structural independence.** Text exists as standalone objects; arrows are dynamic connectors bound to anchors. Moving a module automatically triggers rerouting while preserving topological logic, allowing for seamless human fine-tuning.
>     - **Cons:** While constrained by the primitives of underlying graphics engines, its aesthetic "ceiling" can be significantly raised by encapsulating professional rendering logic into reusable Standardized Skills.
>
> In scientific plotting, **accuracy and editability are paramount.** Therefore, code-based generation holds a decisive advantage. LiveFigure specifically prioritizes this "edit-first" philosophy to meet the rigid requirements of academic publishing.

---

> > ### Author Rebuttal · Reviewer_SLPQ · 2026-04-04
> >
> > Thank you for the detailed rebuttal. I appreciate the authors’ effort in addressing the reviewers’ comments. However, I still find that the technical contribution of the work, particularly in terms of the agentic orchestration and system design, is not yet sufficiently well-justified or clearly distinguished from existing practices. While the proposed system is well-engineered, it remains unclear to what extent it introduces fundamentally new technical insights beyond existing approaches for building similar pipelines. In particular, the paper would benefit from a more explicit characterization of what novel principles or design choices lead to improved capability, rather than relying primarily on system integration. Overall, I believe that these concerns would require a more substantial revision to address, and I will maintain my original score for the current submission.

---

> > > ### Author Response · Authors · 2026-04-04
> > >
> > > We sincerely thank you for your continued engagement and the time you have dedicated to reviewing our work. We would like to take this final opportunity to clarify the core technical contributions of LiveFigure, which differentiate it from a simple engineering integration:
> > >
> > > 1.  **Alignment with the ICML "Applications" Scope: First, we respectfully reiterate that the primary area of our submission is "Applications." According to the official ICML *Call for Papers*, the scope explicitly welcomes _"application-driven machine learning (innovative techniques, problems, and datasets that are of interest to the machine learning community and driven by the needs of end-users)."_ LiveFigure aligns perfectly with this objective by addressing a pervasive daily pain point through principled agentic orchestration, driven directly by the urgent needs of researchers.**
> > > 2. **A Composite Innovation Solving a High-Value Problem:** Second, our innovation is a highly valuable composite one. Identifying and mathematically formulating a previously overlooked but pervasive application problem—generating _natively editable_ scientific vector figures rather than static, unmodifiable raster images—is in itself a crucial contribution to the community. To address this, we proposed a novel agentic orchestration solution that fundamentally bridges the gap between semantic text and structured visual code. As demonstrated in our experiments, this composite approach achieves a significant performance leap, outperforming direct-prompting baselines comprehensively in both executability and layout aesthetics.
> > > 3. **Agentic Workflow Orchestration as a Paramount Innovation Domain:** Finally, with the rapid commoditization of foundational VLMs, the research and design of Agentic Workflows have emerged as a critical domain of technical innovation. Transforming a raw VLM into a reliable, complex workflow is a profoundly non-trivial engineering and algorithmic challenge. In our early explorations, we found that directly relying on VLMs frequently suffers from severe API hallucinations and suboptimal spatial reasoning. Therefore, LiveFigure is far from a simple concatenation of APIs; we introduced essential mechanisms like "Standardized Skills" for cognitive offloading and "Experience-Driven Constraint Injection" to block erroneous reasoning pathways at the source. **In fact, the recent success of pioneering agentic ecosystems (e.g., the OpenClaw project) clearly demonstrates that orchestrating models into functional, robust skills is driving the next wave of AI capabilities. Innovation at the system and agent level should be equally recognized alongside the training of new models.**
> > >
> > > We deeply appreciate your review, which has pushed us to articulate the conceptual core of our work more sharply. We will explicitly include these design principles in the revised manuscript. Thank you again for your constructive efforts to improve our paper.

---

### Decision · Program_Chairs · 2026-04-30

**Decision:**

Accept (regular)

**Comment:**

The paper received four careful reviews including one emergency review.

Reviewer SLPQ considered the problem well motivated but had concerns over the novelty and evaluation. Mostly this was due to the view that the paper makes a systems-level contribution.  They wanted further comparison with SOTA MLLMs like Gemini to justify the contribution.   The reviewer maintains a rejection score (WR) post-rebuttal noting that the latter concerns are addressed but still have the concerns on the novel scope.

Reviewer NswF votes to accept (WA) and maintains the score post-rebuttal, noting all their concerns are addressed.  These primarily relate to clarifications on the metric, the efficiency of the approach and the ablation/attribution of gains.

Reviewer FKAn votes to accept (WA) the paper and notes that their initial concerns are addressed, with their view being reinforced toward acceptance.  The concerns mostly overlapped with NswF.

Reviewer c2sH  raised several queries on the technical method that were addressed in the rebuttal. They maintain their recommendation to accept (WA).


Overall there is majority support to accept this paper.  The main concerns outstanding are from SLPQ but even these are mostly addressed with the remaining one being significance of novel contribution, largely based upon the view this is a systems-level contribution.  I consider the contribution to be sufficient and well justified by the experimental data.  Also in discussion, SLPQ is not objectionable to acceptance given the general view.  Overall the AC recommends the paper is therefore accepted.